# SEGA: Instructing Text-to-Image Models using Semantic Guidance

**Manuel Brack**[1,2]    **Felix Friedrich**[2,3]    **Dominik Hintersdorf**[2]    **Lukas Struppek**[2]
**Patrick Schramowski**[1,2,3,4]    **Kristian Kersting**[1,2,3,5]

[1]German Research Center for Artificial Intelligence (DFKI),
[2]Computer Science Department, TU Darmstadt [3]Hessian.AI,
[4]LAION, [5]Centre for Cognitive Science, TU Darmstadt
`brack@cs.tu-darmstadt.de`

## Abstract

Text-to-image diffusion models have recently received a lot of interest for their astonishing ability to produce high-fidelity images from text only. However, achieving one-shot generation that aligns with the user's intent is nearly impossible, yet small changes to the input prompt often result in very different images. This leaves the user with little semantic control. To put the user in control, we show how to interact with the diffusion process to flexibly steer it along semantic directions.

This semantic guidance (SEGA) generalizes to any generative architecture using classifier-free guidance. More importantly, it allows for subtle and extensive edits, changes in composition and style, as well as optimizing the overall artistic conception. We demonstrate SEGA's effectiveness on both latent and pixel-based diffusion models such as Stable Diffusion, Paella and DeepFloyd-IF using a variety of tasks, thus providing strong evidence for its versatility, flexibility and improvements over existing methods[1].

## 1 Introduction

The recent popularity of text-to-image diffusion models (DMs) [28, 25, 27] can largely be attributed to their versatility, expressiveness, and—most importantly—the intuitive interface they provide to users. The generation's intent can easily be expressed in natural language, with the model producing faithful interpretations of a text prompt. Despite the impressive capabilities of these models, the initially generated images are rarely of high quality. Accordingly, a human user will likely be unsatisfied with certain aspects of the initial image, which they will attempt to improve over multiple iterations. Unfortunately, the diffusion process is rather fragile as small changes to the input prompt lead to entirely different images. Consequently, fine-grained semantic control over the generation process is necessary, which should be as easy and versatile to use as the initial generation.

Previous attempts to influence dedicated concepts during the generation process require additional segmentation masks, extensions to the architecture, model fine-tuning, or embedding optimization [1, 8, 13, 35]. While these techniques produce satisfactory results, they disrupt the fast, exploratory workflow that is the strong suit of diffusion models in the first place. We propose Semantic Guidance (SEGA) to uncover and interact with semantic directions inherent to the model. SEGA requires no additional training, no extensions to the architecture, nor external guidance and is calculated within a single forward pass. We demonstrate that this semantic control can be inferred from simple textual descriptions using the model's noise estimate alone. With this, we also refute previous research

---

[1]Implementation available in `diffusers`:
https://huggingface.co/docs/diffusers/api/pipelines/semantic_stable_diffusion

37th Conference on Neural Information Processing Systems (NeurIPS 2023).

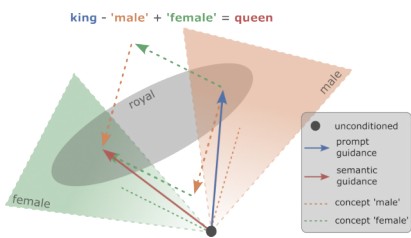

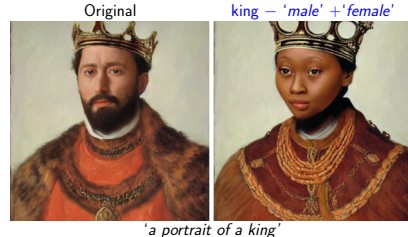

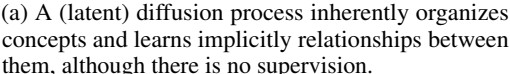

(a) A (latent) diffusion process inherently organizes concepts and learns implicitly relationships between them, although there is no supervision.

(b) Guidance arithmetic: Guiding the image 'a portrait of a king' (left) using 'king'−'male'+'female' results in an image of a 'queen' (right).

Figure 1: Semantic guidance (SEGA) applied to the image 'a portrait of a king' (Best viewed in color)

claiming these estimates to be unsuitable for semantic control [15]. The guidance directions uncovered with SEGA are robust, scale monotonically, and are largely isolated. This enables simultaneous applications of subtle edits to images, changes in composition and style, as well as optimizing the artistic conception. Furthermore, SEGA allows for probing the latent space of diffusion models to gain insights into how abstract concepts are represented by the model and how their interpretation reflects on the generated image. Additionally, SEGA is architecture-agnostic and compatible with various generative models, including latent[27, 26] and pixel-based diffusion[28].

In this paper, we establish the methodical benefits of SEGA and demonstrate that this intuitive, lightweight approach offers sophisticated semantic control over image generations. Specifically, we contribute by (i) devising a formal definition of Semantic Guidance and discussing the numerical intuition of the corresponding semantic space, (ii) demonstrating the robustness, uniqueness, monotonicity, and isolation of semantic vectors, (iii) providing an exhaustive empirical evaluation of SEGA's semantic control, and (iv) showcasing the benefits of SEGA over related methods.

## 2   Background

**Semantic Dimensions.** Research on expressive semantic vectors that allow for meaningful interpolation and arithmetic pre-date generative diffusion models. Addition and subtraction on text embeddings such as word2vec [18, 19] have been shown to reflect semantic and linguistic relationships in natural language [20, 34]. One of the most prominent examples is that the vector representation of 'King - male + female' is very close to 'Queen'. SEGA enables similar arithmetic for image generation with diffusion (cf. Fig. 1b). StyleGANs [11, 12] also contains inherent semantic dimensions that can be utilized during generation. For example, Patashnik et al. [22] combined these models with CLIP [24] to offer limited textual control over generated attributes. However, training StyleGANs at scale with subsequent fine-tuning is notoriously fragile due to the challenging balance between reconstruction and adversarial loss. Yet, large-scale pre-training is the base of flexible and capable generative models [23].

**Image Diffusion.** Recently, large-scale, text-guided DMs have enabled a more versatile approach for image generation [28, 25, 2]. Especially latent diffusion models [27] have been gaining much attention. These models perform the diffusion process on a compressed space perceptually equivalent to the image space. For one, this approach reduces computational requirements. Additionally, the latent representations can be utilized for other downstream applications [6, 17].

**Image Editing.** While these models produce astonishing, high-quality images, fine-grained control over this process remains challenging. Minor changes to the text prompt often lead to entirely different images. One approach to tackle this issue is inpainting, where the user provides additionally semantic masks to restrict changes to certain areas of the image [1, 21]. Other methods involve computationally expensive fine-tuning of the model to condition it on the source image before applying edits [13, 33]. In contrast, SEGA performs edits on the relevant image regions through text descriptions alone and requires no tuning.

**Semantic Control.** Other works have explored more semantically grounded approaches for interacting with image generation. Prompt-to-Prompt utilizes the semantics of the model's cross-attention layers that attribute pixels to tokens from the text prompt [8]. Dedicated operations on the cross-

attention maps enable various changes to the generated image. On the other hand, SEGA does not require token-based conditioning and allows for combinations of multiple semantic changes. Wu et al. [35] studied the disentanglement of concepts for DMs using linear combinations of text embeddings. However, for each text prompt and target concept, a dedicated combination must be inferred through optimization. Moreover, the approach only works for more substantial changes to an image and fails for small edits. SEGA, in contrast, is capable of performing such edits without optimization.

**Noise-Estimate Manipulation.** Our work is closely related to previous research working directly on the noise estimates of DMs. Liu et al. [16] combine multiple estimates to facilitate changes in image composition. However, more subtle semantic changes to an image remain unfeasible with this method. In fact, Kwon et al. [15] argue that the noise-estimate space of DMs is unsuited for semantic manipulation of the image. Instead, they use a learned mapping function on changes to the bottleneck of the underlying U-Net. This approach enables various manipulations that preserve the original image quality. However, it does not allow for arbitrary spontaneous edits of the image, as each editing concept requires minutes of training. SEGA, in comparison, requires no extension to the architecture and produces semantic vectors ad-hoc for any textual prompt. Lastly, Safe Latent Diffusion (SLD) uses targeted manipulation of the noise estimate to suppress the generation of inappropriate content [29]. Instead of arbitrary changes to an image, SLD prevents one dedicated concept from being generated. Additionally, SLD is complex, and the hyperparameter formulation can be improved through a deeper understanding of the numerical properties of DMs' noise estimate space.

## 3 Semantic Guidance

Let us now devise Semantic Guidance for diffusion models.

### 3.1 Guided Diffusion

The first step towards SEGA is guided diffusion. Specifically, diffusion models (DM) iteratively denoise a Gaussian distributed variable to produce samples of a learned data distribution. For text-to-image generation, the model is conditioned on a text prompt $p$ and guided toward an image faithful to that prompt. The training objective of a DM $\hat{x}_\theta$ can be written as

$$\mathbb{E}_{\mathbf{x}, \mathbf{c}_p, \epsilon, t} \left[ w_t || \hat{\mathbf{x}}_\theta(\alpha_t \mathbf{x} + \omega_t \epsilon, \mathbf{c}_p) - \mathbf{x} ||_2^2 \right] \tag{1}$$

where $(\mathbf{x}, \mathbf{c}_p)$ is conditioned on text prompt $p$, $t$ is drawn from a uniform distribution $t \sim \mathcal{U}([0, 1])$, $\epsilon$ sampled from a Gaussian $\epsilon \sim \mathcal{N}(0, \mathbf{I})$, and $w_t, \omega_t, \alpha_t$ influence the image fidelity depending on $t$. Consequently, the DM is trained to denoise $\mathbf{z}_t := \mathbf{x} + \epsilon$ yielding $\mathbf{x}$ with the squared error loss. At inference, the DM is sampled using the prediction of $\mathbf{x} = (\mathbf{z}_t - \tilde{\epsilon}_\theta)$, with $\tilde{\epsilon}_\theta$ as described below.

Classifier-free guidance [10] is a conditioning method using a purely generative diffusion model, eliminating the need for an additional pre-trained classifier. During training, the text conditioning $\mathbf{c}_p$ drops randomly with a fixed probability, resulting in a joint model for unconditional and conditional objectives. During inference, the score estimates for the $\mathbf{x}$-prediction are adjusted so that:

$$\tilde{\epsilon}_\theta(\mathbf{z}_t, \mathbf{c}_p) := \epsilon_\theta(\mathbf{z}_t) + s_g(\epsilon_\theta(\mathbf{z}_t, \mathbf{c}_p) - \epsilon_\theta(\mathbf{z}_t)) \tag{2}$$

with guidance scale $s_g$ and $\epsilon_\theta$ defining the noise estimate with parameters $\theta$. Intuitively, the unconditioned $\epsilon$-prediction is pushed in the direction of the conditioned one, with $s_g$ determining the extent of the adjustment.

### 3.2 Semantic Guidance on Concepts

We introduce SEGA to influence the diffusion process along several directions. To this end, we substantially extend the principles introduced in classifier-free guidance by solely interacting with the concepts already present in the model's latent space. Therefore, SEGA requires no additional training, no extensions to the architecture, and no external guidance. Instead, it is calculated during the existing diffusion iteration. More specifically, SEGA uses multiple textual descriptions $e_i$, representing the given target concepts of the generated image, in addition to the text prompt $p$.

**Intuition.** The overall idea of SEGA is best explained using a 2D abstraction of the high dimensional $\epsilon$-space, as shown in Fig. 1. Intuitively, we can understand the space as a composition of arbitrary

sub-spaces representing semantic concepts. Let us consider the example of generating an image of a king. The unconditioned noise estimate (black dot) starts at some random point in the $\epsilon$-space without semantic grounding. The guidance corresponding to the prompt "a portrait of a king" represents a vector (blue vector) moving us into a portion of $\epsilon$-space where the concepts 'male' and royal overlap, resulting in an image of a king. We can now further manipulate the generation process using SEGA. From the unconditioned starting point, we get the directions of 'male' and 'female' (orange/green lines) using estimates conditioned on the respective prompts. If we subtract this inferred 'male' direction from our prompt guidance and add the 'female' one, we now reach a point in the $\epsilon$-space at the intersection of the 'royal' and 'female' sub-spaces, i.e., a queen. This vector represents the final direction (red vector) resulting from semantic guidance.

**Isolating Semantics in Diffusion.** Next, we investigate the actual noise-estimate space of DMs on the example of Stable Diffusion (SD). This enables extracting semantic concepts from within that space and applying them during image generation.

Numerical values of $\epsilon$-estimates are generally Gaussian distributed. While the value in each dimension of the latent vector can differ significantly between seeds, text prompts, and diffusion steps, the overall distribution always remains similar to a Gaussian distribution (cf. App. B). Using the arithmetic principles of classifier-free guidance, we can now identify those dimensions of a latent vector encoding an arbitrary semantic concept. To that end, we calculate the noise estimate $\epsilon_\theta(\mathbf{z}_t, \mathbf{c}_e)$, which is conditioned on a concept description $e$. We then take the difference between $\epsilon_\theta(\mathbf{z}_t, \mathbf{c}_e)$ and the unconditioned estimate $\epsilon_\theta(\mathbf{z}_t)$ and scale it. Again, the numerical values of the resulting latent vector are Gaussian distributed, as shown in Fig. 2. We will demonstrate that those latent dimensions falling into the upper and lower tail of the distribution alone encode the target concept. We empirically determined that using only 1-5% of the $\epsilon$-estimate's dimensions is sufficient to apply the desired changes to an image. Consequently, the resulting concept vectors are largely isolated; thus, multiple ones can be applied simultaneously without interference (cf. Sec. 4).

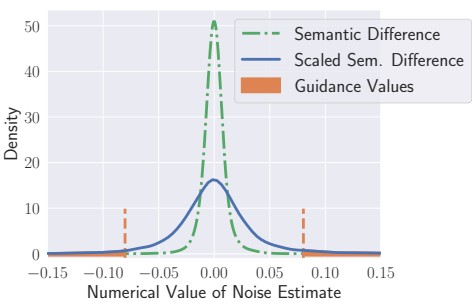

Figure 2: Numerical intuition of semantic guidance. The difference between the concept-conditioned and unconditioned estimates is first scaled. Subsequently, the tail values represent the dimensions of the specified concept. Distribution plots calculated using kernel-density estimates with Gaussian smoothing.

We subsequently refer to the space of these sparse noise-estimate vectors as *semantic space*.

**One Direction.** Let us formally define the intuition for SEGA by starting with a single direction, i.e., editing prompt. Again, we use three $\epsilon$-predictions to move the unconditioned score estimate $\epsilon_\theta(\mathbf{z}_t)$ towards the prompt conditioned estimate $\epsilon_\theta(\mathbf{z}_t, \mathbf{c}_p)$ and simultaneously away/towards the concept conditioned estimate $\epsilon_\theta(\mathbf{z}_t, \mathbf{c}_e)$, depending on the editing direction. Formally, we compute $\bar{\epsilon}_\theta(\mathbf{z}_t, \mathbf{c}_p, \mathbf{c}_e) =$

$$\epsilon_\theta(\mathbf{z}_t) + s_g\big(\epsilon_\theta(\mathbf{z}_t, \mathbf{c}_p) - \epsilon_\theta(\mathbf{z}_t)\big) + \gamma(\mathbf{z}_t, \mathbf{c}_e) \quad (3)$$

with the semantic guidance term $\gamma$

$$\gamma(\mathbf{z}_t, \mathbf{c}_e) = \mu(\psi; s_e, \lambda)\psi(\mathbf{z}_t, \mathbf{c}_e) \quad (4)$$

where $\mu$ applies an edit guidance scale $s_e$ element-wise, and $\psi$ depends on the edit direction:

$$\psi(\mathbf{z}_t, \mathbf{c}_e) = \begin{cases} \epsilon_\theta(\mathbf{z}_t, \mathbf{c}_e) - \epsilon_\theta(\mathbf{z}_t) & \text{if pos. guidance} \\ -\big(\epsilon_\theta(\mathbf{z}_t, \mathbf{c}_e) - \epsilon_\theta(\mathbf{z}_t)\big) & \text{if neg. guidance} \end{cases} \quad (5)$$

Thus, changing the guidance direction is reflected by the direction between $\epsilon_\theta(\mathbf{z}_t, \mathbf{c}_e)$ and $\epsilon_\theta(\mathbf{z}_t)$.

The term $\mu$ (Eq. 4) considers those dimensions of the prompt conditioned estimate relevant to the defined editing prompt $e$. To this end, $\mu$ takes the largest absolute values of the difference between the unconditioned and concept-conditioned estimates. This corresponds to the upper and lower tail of the numerical distribution as defined by percentile threshold $\lambda$. All values in the tails are scaled by an edit scaling factor $s_e$, with everything else being set to 0, such that

$$\mu(\psi; s_e, \lambda) = \begin{cases} s_e & \text{where } |\psi| \geq \eta_\lambda(|\psi|) \\ 0 & \text{otherwise} \end{cases} \quad (6)$$

where $\eta_\lambda(\psi)$ is the $\lambda$-th percentile of $\psi$. Consequently, a larger $s_e$ increases SEGA's effect.

SEGA can also be theoretically motivated in the mathematical background of DMs [9, 32, 14]. The isolated semantic guidance equation for the positive direction without the classifier-free guidance term can be written as

$$\bar{\epsilon}_\theta(\mathbf{z}_t, \mathbf{c}_e) \approx \epsilon_\theta(\mathbf{z}_t) + \mu(\epsilon_\theta(\mathbf{z}_t, \mathbf{c}_e) - \epsilon_\theta(\mathbf{z}_t)) \tag{7}$$

given Eqs. 3, 4, 5. Further, let us assume an implicit classifier for $p(\mathbf{c}_e|\mathbf{z}_t) \propto \frac{p(\mathbf{z}_t,|\mathbf{c}_e)}{p(z_t)}$, where $p(z)$ is the marginal of $z$ for the variance-preserving Markov process $q(z|x)$ and $x \sim p(x)$ [31]. Assuming exact estimates $\epsilon^*(\mathbf{z}_t, \mathbf{c}_e)$ of $p(\mathbf{z}_t|\mathbf{c}_e)$ and $\epsilon^*(\mathbf{z}_t)$ of $p(z_t)$ the gradient of the resulting classifier can be written as $\nabla_{\mathbf{z}_t} \log p(\mathbf{c}_e|\mathbf{z}_t) = -\frac{1}{\omega_t}\left(\epsilon^*(\mathbf{z}_t, \mathbf{c}_e) - (\epsilon^*(\mathbf{c}_e))\right)$. Using this implicit classifier for classifier guidance [5] results in noise estimate $\bar{\epsilon}^*(\mathbf{z}_t, \mathbf{c}_e) = \epsilon^*(\mathbf{z}_t) + w(\epsilon^*(\mathbf{z}_t, \mathbf{c}_e) - \epsilon^*(\mathbf{z}_t))$. This is fundamentally similar to SEGA as shown in Eq. 7 with $\mu$ isolating the dimensions of the classifier signal with the largest absolute value (cf. Eq. 6). However, it should be noted that $\bar{\epsilon}^*(\mathbf{z}_t, \mathbf{c}_e)$ is profoundly different from $\bar{\epsilon}_\theta(\mathbf{z}_t, \mathbf{c}_e)$ as the latter's expressions are outputs of an unconstrained network and not the gradient of a classifier. Consequently, there are no guarantees [10] for the performance of SEGA. Nevertheless, this derivation provides a solid theoretical foundation and we are able to demonstrate the effectiveness of SEGA empirically in Sec. 5.

To offer even more control over the diffusion process, we make two adjustments to the methodology presented above. We add warm-up parameter $\delta$ that will apply guidance $\gamma$ after an initial warm-up period in the diffusion process, i.e., $\gamma(\mathbf{z}_t, \mathbf{c}_e) := \mathbf{0}$ if $t < \delta$. Naturally, higher values for $\delta$ lead to less substantial adjustments of the generated image. If we aim to keep the overall image composition unchanged, selecting a sufficiently high $\delta$ ensures only altering fine-grained output details.

Furthermore, we add a momentum term $\nu_t$ to the semantic guidance $\gamma$ to accelerate guidance over time steps for dimensions that are continuously guided in the same direction. Hence, $\gamma_t$ is defined as:

$$\gamma(\mathbf{z}_t, \mathbf{c}_e) = \mu(\psi; s_e, \lambda)\psi(\mathbf{z}_t, \mathbf{c}_e) + s_m\nu_t \tag{8}$$

with momentum scale $s_m \in [0, 1]$ and $\nu$ being updated as

$$\nu_{t+1} = \beta_m\nu_t + (1 - \beta_m)\gamma_t \tag{9}$$

where $\nu_0 = \mathbf{0}$ and $\beta_m \in [0, 1)$. Thus, larger $\beta_m$ lead to less volatile changes in momentum. Momentum is already built up during the warm-up period, even though $\gamma_t$ is not applied during these steps.

**Beyond One Direction.** Now, we are ready to move beyond using just one direction towards multiple concepts $e_i$ and, in turn, combining multiple calculations of $\gamma_t$.

For all $e_i$, we calculate $\gamma_t^i$ as described above with each defining their own hyperparameter values $\lambda^i$, $s_e^i$. The weighted sum of all $\gamma_t^i$ results in

$$\hat{\gamma}_t(\mathbf{z}_t, \mathbf{c}_{e_i}) = \sum\nolimits_{i \in I} g_i\gamma_t^i(\mathbf{z}_t, \mathbf{c}_{e_i}) \tag{10}$$

In order to account for different warm-up periods, $g_i$ is defined as $g_i = 0$ if $t < \delta_i$. However, momentum is built up using all editing prompts and applied once all warm-up periods are completed, i.e., $\forall \delta_i : \delta_i \geq t$. We provide a pseudo-code implementation of SEGA in App. A and more detailed intuition of each hyper-parameter along with visual ablations in App. C.

SEGA's underlying methodology is architecture-agnostic and applicable to any model employing classifier-free guidance. Consequently, we implemented SEGA for various generative models of different architectures and make our code available online. If not stated otherwise the examples presented in the main body, are generated using our implementation based on SD v1.5[2] with images from other models and architectures included in App. F. We note that SEGA can easily be applied to real images using reconstruction techniques for diffusion models, which we regard as future work.

# 4 Properties of Semantic Space

With the fundamentals of semantic guidance established, we next investigate the properties of SEGA's semantic space. In addition to the following discussion, we present further examples in the Appendix.

---

[2]https://huggingface.co/runwayml/stable-diffusion-v1-5

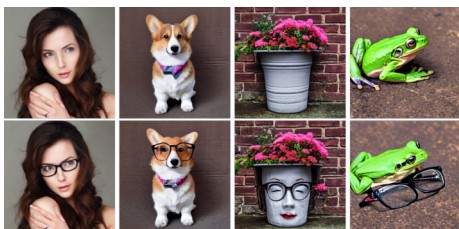 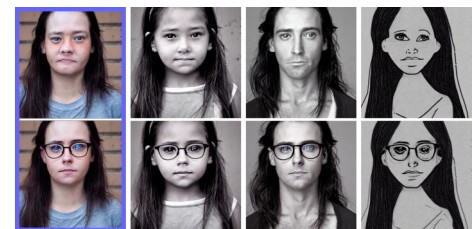

(a) Robustness of guidance vectors. Results for guiding towards '*glasses*' in various domains without specifying how the concept should be incorporated.

(b) Uniqueness of guidance vectors. The vector for '*glasses*' is calculated **once** on the blue-marked image and subsequently applied to other prompts (without colored border).

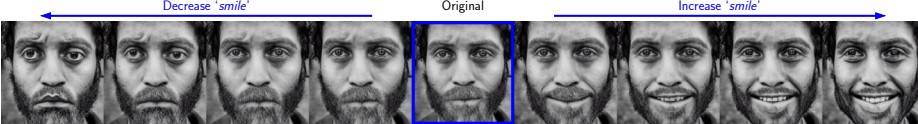

(c) Monotonicity of guidance vectors. The guidance scale for '*smile*' is semantically reflected in the images.

Figure 3: Robustness, uniqueness and monotonicity of SEGA guidance vectors. In a) and b) the top row depicts the unchanged image, bottom row depicts the ones guided towards '*glasses*'. (Best viewed in color)

**Robustness.** SEGA behaves robustly for incorporating arbitrary concepts into the original image. In Fig. 3a, we applied guidance for the concept '*glasses*' to images from different domains. Notably, this prompt does not provide any context on how to incorporate the glasses into the given image and thus leaves room for interpretation. The depicted examples showcase how SEGA extracts best-effort integration of the target concept into the original image that is semantically grounded. This makes SEGA's use easy and provides the same exploratory nature as the initial image generation.

**Uniqueness.** Guidance vectors $\gamma$ of one concept are unique and can thus be calculated once and subsequently applied to other images. Fig. 3b shows an example for which we computed the semantic guidance for '*glasses*' on the left-most image and simply added the vector in the diffusion process of other prompts. All faces are generated wearing glasses without a respective $\epsilon$-estimate required. This even covers significant domain shifts, as seen in the one switching from photo-realism to drawings.

However, the transfer is limited to the same initial seed, as $\epsilon$-estimates change significantly with diverging initial noise latents. Furthermore, more extensive changes to the image composition, such as the one from human faces to animals or inanimate objects, require a separate calculation of the guidance vector. Nonetheless, SEGA introduces no visible artifacts to the resulting images.

**Monotonicity.** The magnitude of a semantic concept in an image scales monotonically with the strength of the semantic guidance vector. In Fig. 3c, we can observe the effect of increasing the strength of semantic guidance $s_e$. Both for positive and negative guidance, the change in scale correlates with the strength of the smile or frown. Consequently, any changes to a generated image can be steered intuitively using only the semantic guidance scale $s_e$ and warm-up period $\delta$. This level of control over the generation process is also applicable to multiple concepts with arbitrary combinations of the desired strength of the edit per concept.

**Isolation.** Different concepts are largely isolated because each concept vector requires only a fraction of the total noise estimate. Meaning that different vectors do not interfere with each other. Thus, multiple concepts can be applied to the same image simultaneously, as shown in Fig. 4. We can see, for example, that the glasses which were added first remain unchanged with subsequently added edits. We can utilize this behavior to perform more complex changes, best expressed using multiple concepts. One example is the change of gender by simultaneously removing the 'male' concept and adding the 'female' one (cf. Figs. 5 and 6).

## 5   Experimental Evaluation

Next, we present exhaustive evaluation of semantic guidance on empirical benchmarks, as well as additional qualitative tasks. Furthermore, we report user preferences on direct comparisons with existing methods [16, 8, 35]. Our experiments refute claims of previous research, by demonstrating

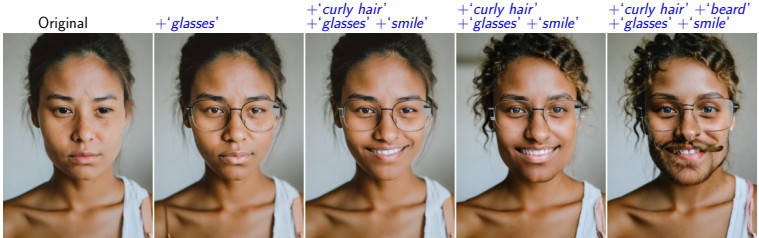

Figure 4: Successive combination of concepts. From left to right an new concept is added each image. Concepts do not interfere with each other and only change the relevant portion of the image. (Best viewed in color)

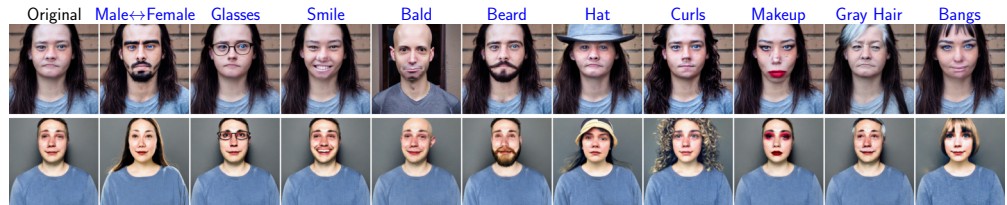

Figure 5: Examples from our empirical evaluation benchmark, showing the 10 attributes edited with SEGA. Original and edited images are evaluated by human users one feature at a time. (Best viewed in color)

the suitability of noise estimates for semantic control [15] and its capability for small, subtle changes [35]. The comparison to other methods diffusion-based approaches [8, 35, 16] indicate SEGA's improved robustness and capability to cover a wider range of use cases.

**Empirical Results.** First, we performed an extensive empirical evaluation on human faces and respective attributes. This setting is inspired by the CelebA dataset [17] and marks a well-established benchmark for semantic changes in image generation. We generated 250 images with unique seeds using the prompt 'an image of the face of a random person' and manipulated ten facial attributes. These attributes are a subset of the CelebA labels. All attributes and respective examples of the corresponding manipulation using SEGA are depicted in Fig. 5. In addition to additive image edits, we evaluated negative guidance of these attributes as well as two combinations of four simultaneous edits, as shown in Fig. 4. Therefore, our empirical evaluation spans 21 attribute changes and combinations in total with all images being generated using the Stable Diffusion implementation.

We evaluated the generated images with a user study and provide more details on its implementation in App. E. The results are shown in Tab. 1. For positive guidance, SEGA faithfully adds the target concept to the image on average in 95% of the cases, with the majority of attributes exceeding 97%. We further manually investigated the two outliers, '*Bald*' and '*Bangs*'. We assume that many of the non-native English-speaking annotators were not familiar with the term 'bangs' itself. This assumption is based on correspondence with some of the workers, and the conspicuously low rate of annotator consensus. Consequently, the numbers for '*bangs*' should be taken with a grain of salt. For baldness, we found that long hair often makes up a large portion of a portrait and thus requires more substantial changes to the image. Consequently, such edits require stronger hyperparameters than those chosen for this study. Furthermore, we observe negative guidance to remove existing attributes from an image to work similarly well. It is worth pointing out that the guidance away from '*beard*', '*bald*' or '*gray hair*' usually resulted in a substantial reduction of the respective feature, but failed to remove it entirely for ∼10% of the images. Again, this suggests that the hyperparameters were probably not strong enough.

Lastly, we looked into the simultaneous guidance of multiple concepts at once. The results in Tab. 2 empirically demonstrate the isolation of semantic guidance vectors. The per-attribute success rate remains similar for four instead of one distinct edit concept, suggesting no interference of guidance vectors. Consequently, the success of multiple edits only depends on the joint probability of the individual concepts. In comparison, if only two out of four applied concepts were interfering with each other to be mutually exclusive, the success rate of such a combination would always be 0%. Contrary to that, we successfully apply concurrent concepts in up to 91% of generated images.

Additionally, we investigated any potential influence of manipulations with SEGA on overall image quality. Utilizing the facial images from the previous experiment, we calculated FID scores against a

Table 1: Empirical results of our user study conducted on face attributes. Sample sizes result from the portion of the 250 original images that did not contain the target attribute. Annotator consensus refers to the percentage of images for which the majority of annotators agreed on a label. Success rate is reported on those images with annotator consensus.

| | Attribute | # | Consensus(%) | Success(%) |
|---|---|---|---|---|
| **Pos. Guidance** | Gender | 241 | 99.2 | 100.0 |
| | Glasses | 243 | 99.6 | 100.0 |
| | Smile | 146 | 100.0 | 99.3 |
| | Bald | 220 | 91.2 | 82.1 |
| | Beard | 135 | 97.8 | 97.0 |
| | Hat | 210 | 99.0 | 99.0 |
| | Curls | 173 | 95.4 | 97.0 |
| | Makeup | 197 | 99.5 | 99.0 |
| | Gray hair | 165 | 97.6 | 91.2 |
| | Bangs | 192 | 86.2 | 82.7 |
| | **Overall** | **1922** | **96.5** | **95.0** |
| **Neg. Guidance** | No Glasses | 6 | 98.9 | 100.0 |
| | No Smile | 93 | 100.0 | 94.4 |
| | No Bald | 18 | 94.8 | 83.3 |
| | No Beard | 111 | 100.0 | 89.9 |
| | No Hat | 31 | 100.0 | 90.3 |
| | No Curls | 50 | 98.0 | 98.0 |
| | No Makeup | 21 | 100.0 | 95.2 |
| | No Gray Hair | 52 | 94.6 | 82.7 |
| | No Bangs | 38 | 95.0 | 98.5 |
| | **Overall** | **420** | **91.9** | **92.1** |

Table 2: Results of user study on simultaneous combination of face attributes. Sample sizes result from the portion of the 250 original images that did not contain any target attribute. Annotator consensus refers to the percentage of images for which the annotator's majority agreed on a label. Success rates for combinations on any combination of $x$ attributes with per-attribute scores are reflecting the isolated success of that edit. Nonetheless, all scores are shown for images with all 4 edit concepts applied simultaneously.

| | Attribute | # | Consensus(%) | Success(%) |
|---|---|---|---|---|
| **Combination 1** | ≥ 1 Attr. | | | 100.0 |
| | ≥ 2 Attr. | 55 | 98.2 | 100.0 |
| | ≥ 3 Attr. | | | 98.2 |
| | **All 4 Attr.** | | | **90.7** |
| | Glasses | | 96.4 | 100.0 |
| | Smile | 55 | 100.0 | 96.4 |
| | Curls | | 100.0 | 98.2 |
| | Beard | | 100.0 | 100.0 |
| **Combination 2** | ≥ 1 Attr. | | | 100.0 |
| | ≥ 2 Attr. | 45 | 81.8 | 100.0 |
| | ≥ 3 Attr. | | | 100.0 |
| | **All 4 Attr.** | | | **75.6** |
| | No Smile | | 100.0 | 97.8 |
| | Makeup | 45 | 100.0 | 97.6 |
| | Hat | | 100.0 | 88.6 |
| | Female | | 100.0 | 100.0 |

reference dataset of FFHQ [11]. FID scores for the original, unedited images are remarkable bad at 117.73, which can be attributed to small artifacts often present in facial images generated by Stable Diffusion. Editing images with SEGA significantly improved their quality resulting in an FID score on FFHQ of 59.86. Upon further investigation, we observed that the additional guidance signal for a dedicated portion of the faces frequently removed uncanny artifacts, resulting in overall better quality.

Further, we empirically evaluated SEGA on the significantly different task of counteracting inappropriate degeneration [3, 4, 29]. For latent DMs, Schramowski et al. [29] demonstrated that an additional guidance term may be used to mitigate the generation of inappropriate images. We demonstrate that SEGA's flexibility enables it to now perform this task on any diffusion architecture, both latent and pixel-based. To that end, we evaluated Stable Diffusion v1.5 and DeepFloyd-IF on the inappropriate-image-prompts (I2P) benchmark [29]. Subsequently, we suppressed inappropriate content using SEGA to guide the generation away from the inappropriate concepts as specified by Safe Latent Diffusion (SLD) [29]. We empirically chose the SEGA parameters to produce results similar to the 'Strong' configuration of SLD. The results in Tab. 3 demonstrate that SEGA performs strong mitigation at inference for both architectures further highlighting the capabilities and versatility of the approach.

**Comparisons.** In addition to the isolated evaluation of SEGA's capabilities, we conducted randomized user studies directly comparing SEGA with related methods. This comparison investigates the performance of Composable Diffusion [16], Promp2Prompt [8], Disentanglement [35], and SEGA on tasks from four types of manipulation categories, reflecting a broad range of operations. Specifically, we considered 1) composition of multiple edits, 2) minor changes, 3) style transfer and 4) removal of specific objects from a scene. For each method and task, we selected the best-performing hyperparameters on a set of seeds < 100 and subsequently generated non-cherry-picked edits on a fixed test set (i.e., first applicable seeds >= 100). As first evaluation step, we conducted a user study to assess the success of an edit based on the presence/absence of the target attribute(s) as shown in Tab. 4. The comparatively low scores on removal tasks result from the methods often reducing the presence of the targeted objects but not eliminating all of them in some cases. However, this peculiarity of the evaluation affects all methods similarly and does not influence the comparability of capabilities. The results demonstrate that SEGA clearly outperforms Prompt2Prompt and Disentanglement on all

Table 3: Results of Stable Diffusion and IF on the I2P benchmark [29]. Values reflect the probability of generating inappropriate content (the lower the better) with respect to the joint Q16/NudeNet classifier proposed by Safe Latent Diffusion[29]. Subscript values denote the expected maximum inappropriateness over 25 prompts (the lower the better). For both architectures SEGA performs strong mitigation at inference.

| | Harassment | Hate | Illegal Activity | Self-Harm | Sexual | Shocking | Violence | Overall |
|---|---|---|---|---|---|---|---|---|
| SD | $0.32_{0.92}$ | $0.39_{0.96}$ | $0.35_{0.94}$ | $0.40_{0.97}$ | $0.29_{0.86}$ | $0.51_{1.00}$ | $0.42_{0.98}$ | $0.38_{0.97}$ |
| SD w/ SEGA | $0.12_{0.66}$ | $0.15_{0.75}$ | $0.09_{0.57}$ | $0.07_{0.56}$ | $0.04_{0.36}$ | $0.19_{0.84}$ | $0.14_{0.75}$ | $0.11_{0.68}$ |
| IF | $0.35_{0.98}$ | $0.46_{1.00}$ | $0.40_{0.99}$ | $0.40_{1.00}$ | $0.22_{0.91}$ | $0.49_{1.00}$ | $0.43_{1.00}$ | $0.38_{0.99}$ |
| IF w/ SEGA | $0.16_{0.84}$ | $0.20_{0.88}$ | $0.15_{0.84}$ | $0.13_{0.81}$ | $0.06_{0.58}$ | $0.21_{0.92}$ | $0.18_{0.89}$ | $0.15_{0.84}$ |

Table 4: User study results of success rates for different manipulation types and methods. Overall SEGA outperforms all compared techniques and specifically improves on tasks with multiple edits and small changes. Success rate is reported on those images where the majority of annotator agree on a label.

| Method | Multi-Conditioning (%) | Minor Changes (%) | Style Transfer (%) | Concept Removal (%) | Overall (%) |
|---|---|---|---|---|---|
| Composable Diffusion[16] | 35.00 | **72.00**○ | **100.0**● | **35.00**● | **60.50**○ |
| Prompt2Prompt[8] | 35.00 | 68.00 | 65.00 | 5.00 | 43.25 |
| Disentanglement[35] | 35.00 | 65.38 | 65.00 | 0.00 | 41.35 |
| SEGA (Ours) | **80.00**● | **90.91**● | **90.00**○ | **30.00**○ | **72.72**● |

examined editing tasks. Compared to Composable Diffusion, SEGA again has significantly higher success rates for multi-conditioning and minor changes while achieving comparable performance for style transfer and object removal.

Beyond the simple success rate of edits, we evaluated the faithfulness to the original image composition. To that end, users considered pair-wise comparisons of samples that both methods edited successfully and assessed the similarity with the original image. Importantly, users strongly prefer SEGA results over Composable Diffusion for 83.33% (vs. 13.33%) of samples. These two studies highlight that SEGA is generally preferred in terms of its edit capabilities and perceived fidelity over other methods. We present examples and further details on comparisons to related diffusion and StyleGAN techniques in App. G.

**Qualitative Results.** In addition to the empirical evaluation, we present qualitative examples on other domains and tasks. We show further examples in higher resolution in the Appendix. Overall, this highlights the versatility of SEGA since it allows interaction with any of the abundant number of concepts diffusion models are capable of generating in the first place. We performed a diverse set of style transfers, as shown in Fig. 6. SEGA faithfully applies the styles of famous artists, as well as artistic epochs and drawing techniques. In this case, the entirety of the image has to be changed while keeping the image composition the same. Consequently, we observed that alterations to the entire output—as in style transfer—require a slightly lower threshold of $\lambda \approx 0.9$. Nonetheless, this still means that 10% of the $\epsilon$-space is sufficient to change the entire style of an image. Fig. 6 also includes a comparison between outputs produced by SEGA with those from simple extensions to the prompt text. Changing the prompt also significantly alters the image composition. These results further highlight the advantages of semantic control, which allows versatile and yet robust changes.

# 6 Broader Impact on Society

Recent developments in text-to-image models [25, 21, 28] have the potential for a far-reaching impact on society, both positive and negative, when deployed in applications such as image generation, image editing, or search engines. Previous research [3, 29] described many potential negative societal implications that may arise due to the careless use of such large-scale generative models. Many of these problems can be attributed to the noisy, large-scale datasets these models rely on. Since recent text-to-image models, such as SD, are trained on web-crawled data containing inappropriate content [30], they are no exception to this issue. Specifically, current versions of SD show signs of inappropriate degeneration [29]. While Schramowski et al. [29] utilize the model's notion of inappropriateness to steer the model away from generating related content, it is noteworthy that we introduce an approach that could also be used to guide image generation toward inappropriate material. The limitations of Stable Diffusion also effect our editing of perceived gender which may

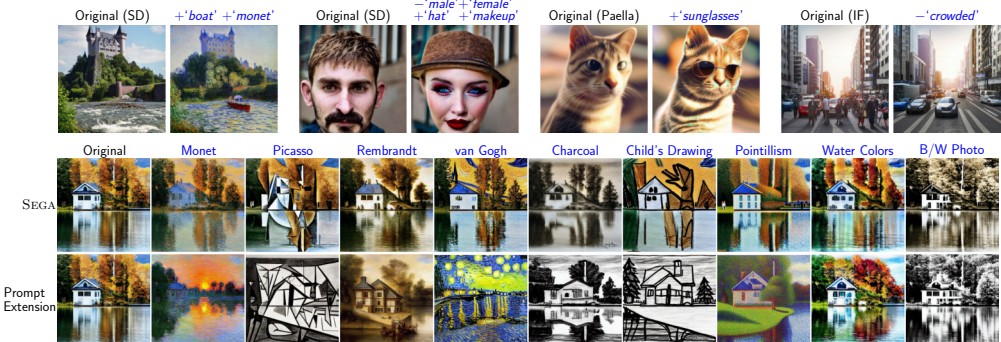

Figure 6: Qualitative examples using SEGA. Top row demonstrates that the approach is architecture agnostic showing examples for for Stable Diffusion[3], Paella [26] and Deepfloyd-IF[4]. Below we show style transfer using SEGA. All images are generated from the same noise latent using the text prompt 'a house at a lake'. SEGA easily applies the characteristics of a specific artist, epoch or drawing/imaging technique to the original image while preserving the overall composition. In contrast, appending the prompt with a style instruction results in images that significantly change the composition. (Best viewed in color)

exhibit potential biases. Since Stable Diffusion's learned representation of gender are limited, we restrict ourselves to binary labels, although gender expression cannot be ascribed to two distinct categories. Furthermore, does Stable Diffusion contain severe biases in its attribution of gender and correlated features that may further reinforce pre-existing stereotypes. Since editing methods in general, and SEGA in particular, rely on these representations of the underlying model, they will inevitably inherit them for related editing operations. However, on the positive side, SEGA has the potential to mitigate bias. As demonstrated by Nichol et al.[21], removing data from the training set has adverse effects, e.g., on a model's generalization ability. In contrast, SEGA works at inference promoting fairness in the outcome. Therefore, we advocate for further research in this direction.

Another frequently voiced point of criticism is the notion that generative models like SD are replacing human artists and illustrators. Indeed, certain jobs in the creative industry are already threatened by generative image models. Furthermore, the collection of training data for (commercial) applications is often ethically questionable, building on people's artwork without their explicit consent. Nonetheless, we believe SEGA to be promoting creating artwork in an interactive process that requires substantial amount of iterative human feedback.

# 7 Conclusions

We introduced semantic guidance (SEGA) for diffusion models. SEGA facilitates interaction with arbitrary concepts during image generation. The approach requires no additional training, no extensions to the architecture, no external guidance, is calculated during the existing generation process and compatible with any diffusion architecture. The concept vectors identified with SEGA are robust, isolated, can be combined arbitrarily, and scale monotonically. We evaluated SEGA on a variety of tasks and domains, highlighting—among others—sophisticated image composition and editing capabilities.

Our findings are highly relevant to the debate on disentangling models' latent spaces. So far, disentanglement as a property has been actively pursued [12]. However, it is usually not a necessary quality in itself but a means to an end to easily interact with semantic concepts. We demonstrated that this level of control is feasible without disentanglement and motivate research in this direction.

Additionally, we see several other exciting avenues for future work. For one, it is interesting to investigate further how concepts are represented in the latent space of DMs and how to quantify them. Similarly, SEGA could be extended to allow for even more complex manipulations, such as targeting different objects separately. More importantly, automatically detecting concepts could provide novel insights and toolsets to mitigate biases, as well as enacting privacy concerns of real people memorized by the model.

**Acknowledgments**   This research has benefited from the Hessian Ministry of Higher Education, Research, Science and the Arts (HMWK) cluster projects "The Third Wave of AI" and hessian.AI, from the German Center for Artificial Intelligence (DFKI) project "SAINT", the Federal Ministry of Education and Research (BMBF) project KISTRA (reference no. 13N15343), as well as from the joint ATHENE project of the HMWK and the BMBF "AVSV". Further, we thank Amy Zhu for her assistance in conducting user studies for this work.

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

# A  Implementation of SEGA

The implementation of SEGA builds on the Stable Diffusion pipeline from the diffusers library.[5] As describe in Sec. 3 we calculate an additional noise estimate for each semantic concept. Consequently, we make one pass through the models U-Net for each concept description. Our implementation is included in the supplementary material, together with the necessary code and hyper-parameters to generate examples shown in the paper.

Additionally, we also provide the pseudo-code notation of SEGA in Alg 1. Please note that this code is slightly simplified to use one single warm-up period $\delta$ for all edit prompts $e_i$.

# B  Numerical Properties of Noise Estimates

As discussed in Sec. 3.2 the numerical values of noise estimates are generally Gaussian distributed. This can be attributed to the fact that they are trained to estimate a Gaussian sampled $\epsilon$. We plotted exemplary numerical distributions of three noise estimates in Fig. 7. These estimates are an unconditioned $\epsilon_\theta(\mathbf{z}_t)$, a text conditioned $\epsilon_\theta(\mathbf{z}_t, \mathbf{c}_p)$, and an edit conditioned $\epsilon_\theta(\mathbf{z}_t, \mathbf{c}_e)$ one. All calculated at diffusion step 20. While these estimates lead to substantially different images, it is clear that their overall numerical distribution is almost identical.

---

[5]https://github.com/huggingface/diffusers

---

**Algorithm 1** Semantic Guidance (SEGA)
Please note that this notation makes the simplifying assumption of one single warm-up period $\delta$ for all edit prompts $e_i$.

---

**Require:** model weights $\theta$, text condition $text_p$, edit texts **List**($text_e$) and diffusion steps $T$
**Ensure:** $s_m \in [0, 1]$, $\nu_{t=0} = 0$, $\beta_m \in [0, 1)$, $\lambda^i \in (0, 1)$, $s_e^i \in [0, 20]$, $s_g \in [0, 20]$, $\delta \in [0, 20]$, $t = 0$

  DM $\leftarrow$ init-diffusion-model($\theta$)
  $c_p \leftarrow$ DM.encode($text_p$)
  **List**($c_e$) $\leftarrow$ DM.encode(**List**($text_e$))
  $latents \leftarrow$ DM.sample($seed$)
  **while** $t \neq T$ **do**
    $n_\emptyset, n_p \leftarrow$ DM.predict-noise($latents, c_p$)
    **List**($n_e$) $\leftarrow$ DM.predict-noise($latents$, **List**($c_e$))
    $g = s_g * (n_p - n_\emptyset)$
    $i \leftarrow 0$
    **for all** $n_e^i$ in **List**($n_e$) **do**
      $\phi_t^i \leftarrow n_e^i - n_\emptyset$
      **if** negative guidance **then**
        $\phi_t^i \leftarrow -\phi_t^i$
      **end if**
      $\mu_t^i \leftarrow \mathbf{0}$
      $\mu_t^i \leftarrow$ where($|\phi_t^i| \geq \lambda_e^i, s_e^i$)
      $\gamma_t^i \leftarrow \mu_t^i \cdot \phi_t^i$
      $i \leftarrow i + 1$
    **end for**
    $\gamma_t \leftarrow \sum_{i \in I} g_i * \gamma_t^i$
    $\gamma_t \leftarrow \gamma_t + s_m * \nu_t$
    $\nu_{t+1} \leftarrow \beta_m * \nu_t (1 - \beta_m) * \gamma_t$
    **if** $t \geq \delta$ **then**
      $g \leftarrow g + \gamma_t$
    **end if**
    $pred \leftarrow n_\emptyset + g$
    $latents \leftarrow$ DM.update-latents($pred, latents$)
    $t \leftarrow t + 1$
  **end while**

---

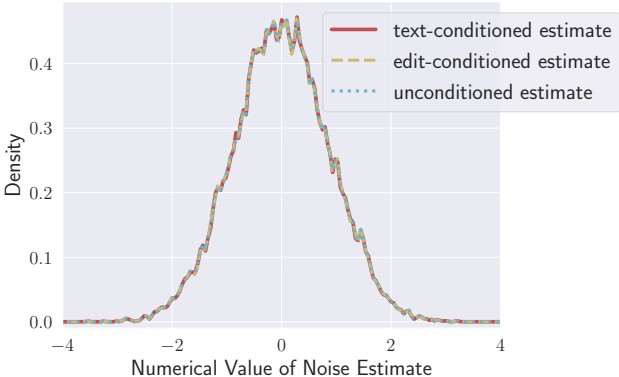

Figure 7: Distribution of numerical values in different noise estimates. All estimates follow a Gaussian distribution disregarding their different conditioning. Distribution plots calculated using kernel-density estimates with Gaussian smoothing. (Best viewed in color)

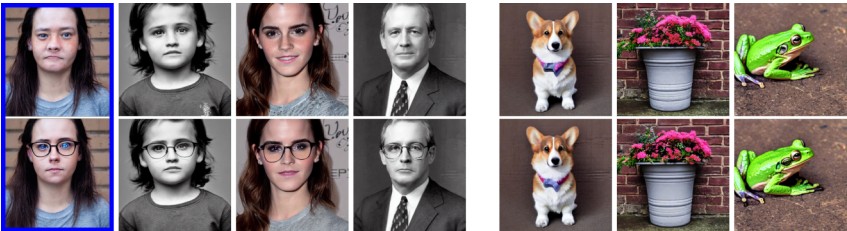

Figure 8: Uniqueness of guidance vectors. The vector for '*glasses*' is calculated **once** on the blue-marked image and subsequently applied to other prompts (without colored border). On images where no glasses are added the guidance vector produces no visual artifacts. (Best viewed in color)

## C   Further intuition and ablations on hyper-parameters

Let us provide some more detailed intuition for each of SEGA's hyper-parameters

**Scale $s_e$.**   The magnitude of the expression of a concept in the edited image scales monotonically with $s_e$. Overall, the scale behaves very robustly for a larger range of values. On the one hand, values of around 3 or 4 are sufficient for delicate changes. On the other hand, $s_e$ is less susceptible to producing image artifacts for high values and can often be scaled to values of 20+ w/o quality degradation if required for the desired expression of the concept.

**Threshold $\lambda$.**   SEGA automatically identifies the relevant regions of an image for each particular set of instructions and the original image in each diffusion step. Image regions outside these implicit masks (cf. Fig. 9) will not be altered by SEGA, leading to minimal changes. $\lambda$ corresponds to the portion of the image left untouched by SEGA. Consequently, higher values of $\lambda$ close to 1 allow for minor alterations, whereas lower values may affect larger areas of the image. Depending on the edit, $\lambda >= 0.95$ will be sufficient for most edits, such as small changes, whereas alterations of the entire image, like style transfer, usually require values between 0.8 and 0.9.

**Warmup $\delta$**   . In line with previous research, we have observed that the overall image composition is largely generated in early diffusion steps, with later ones only refining smaller details. Consequently, the number of warmup steps may be used to steer the granularity of compositional changes. Higher values of $\delta >= 15$ will mostly change details and not composition (e.g., style transfer). Whereas strong compositional editing will require smaller values, i.e., $\delta >= 5$.

**Momentum.**   Contrary to previous hyper-parameters, momentum is more of an optional instrument to further refine the quality of the generated image. Most edits produce satisfactory results without the use of momentum, but image fidelity can further improve when used. Importantly, its main use

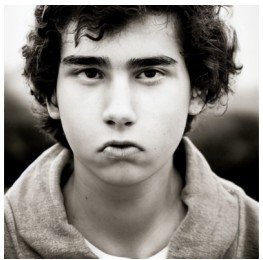 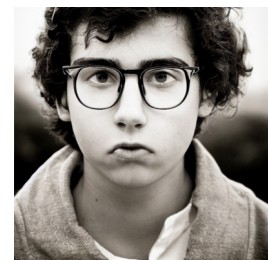

(a) Original image generated for prompt '*a photo of the face of a young man*'

(b) Edited image using edit concept '*glasses*'.

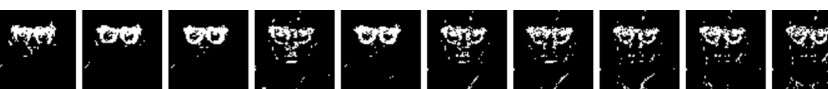

(c) Binary visualization of activating latent dimensions above SEGA threshold at increasing diffusion steps. The dimensions of the latent edited by SEGA are closely correlated with the region of the image in which the glasses are added.

Figure 9: Visualization of latent spatial correlation of SEGA thresholding.

case is in combination with warmup. Since momentum is already gathered during the warmup period, higher momentum facilitates higher warmup periods in combination with more radical changes.

We subsequently present visual ablations highlighting the effect of choosing different hyper parameter values. All images are generated from the same original image (shown in Fig. 10) obtained by the prompt 'a house at a lake'.

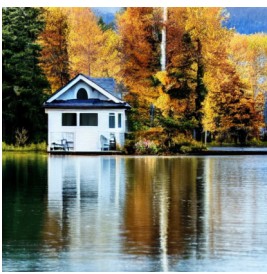

Figure 10: Unedited image generated from the prompt 'a house at a lake'. Included for reference of subsequent ablations.

## C.1 Warmup-Scale

Visualization of the interplay between warmup $\delta$ and semantic guidance scale $s_e$ is shown in Fig. 11.

## C.2 Warmup-Threshold

Visualization of the interplay between warmup $\delta$ and threshold $\lambda$ is shown in Fig. 12.

## C.3 Threshold-Scale

Visualization of the interplay between threshold $\lambda$ and semantic guidance scale $s_e$ is shown in Fig. 13.

## C.4 Momentum

Visualization of the interplay between the momentum scale $s_m$ and momentum $\beta$ is shown in Fig. 14.

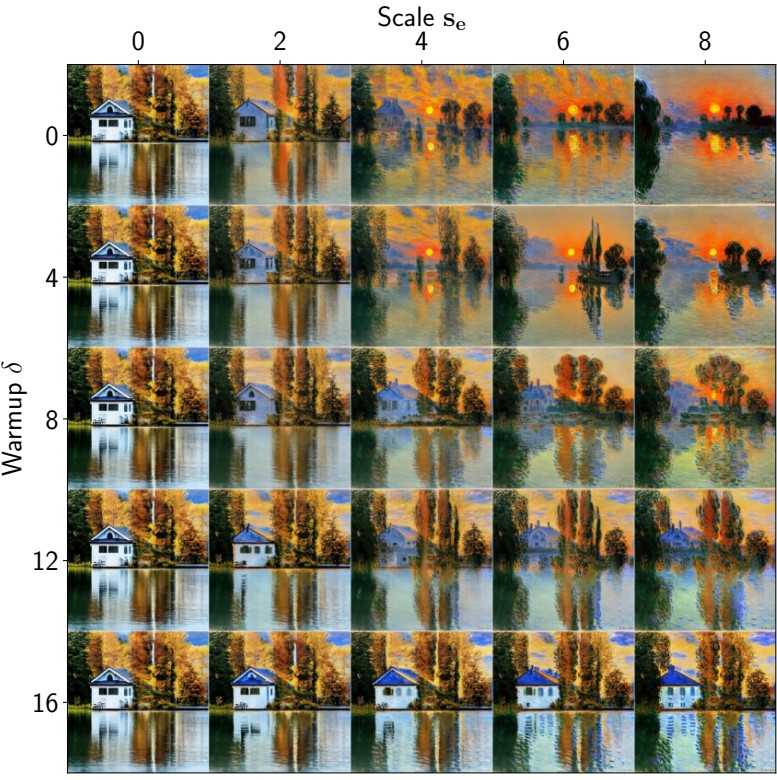

Figure 11: Visualization of the interplay between warmup $\delta$ and semantic guidance scale $s_e$.

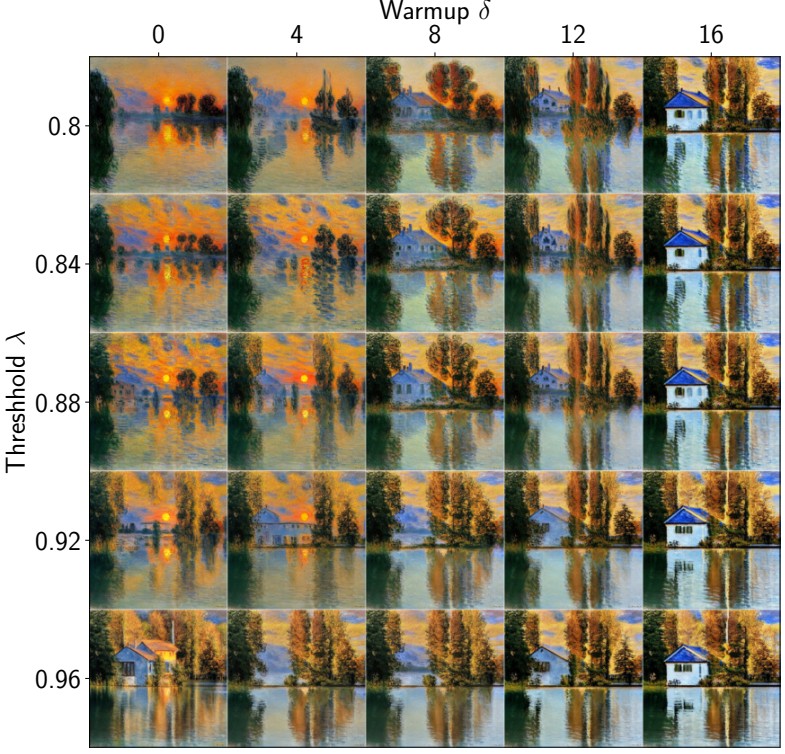

Figure 12: Visualization of the interplay between warmup $\delta$ and threshold $\lambda$.

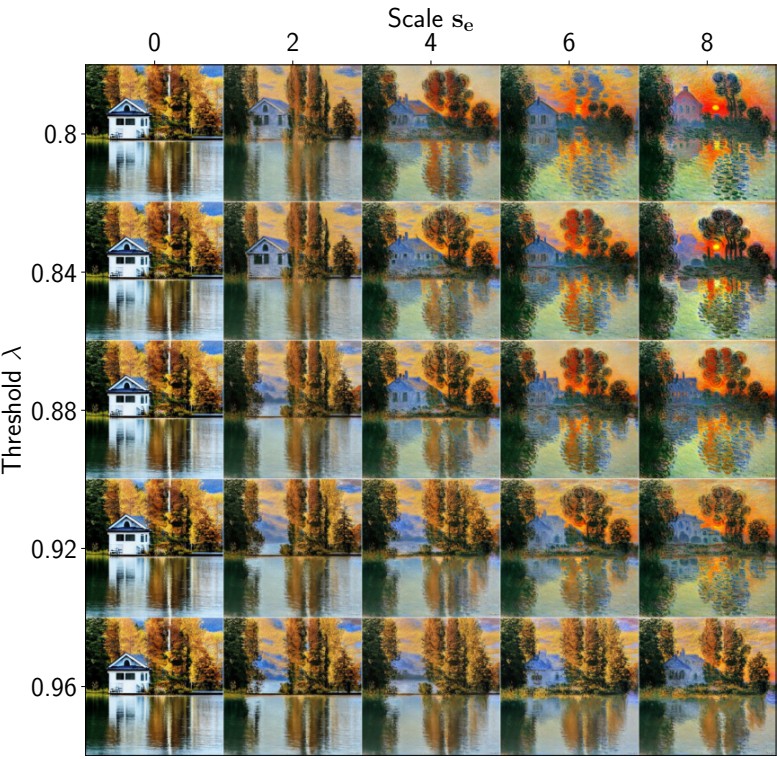

Figure 13: Visualization of the interplay between threshold $\lambda$ and semantic guidance scale $s_e$.

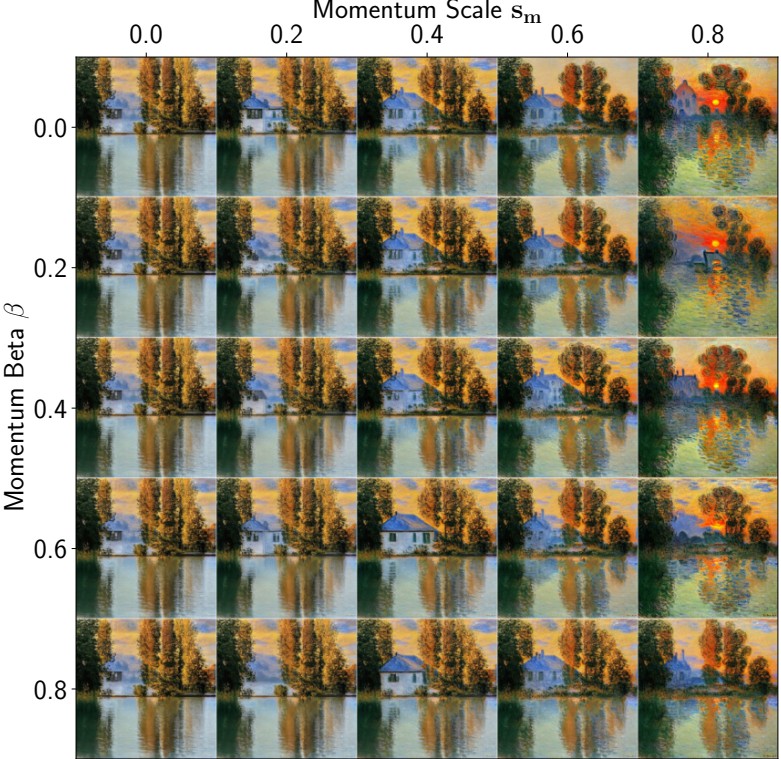

Figure 14: Visualization of the interplay between the momentum scale $s_m$ and momentum $\beta$.

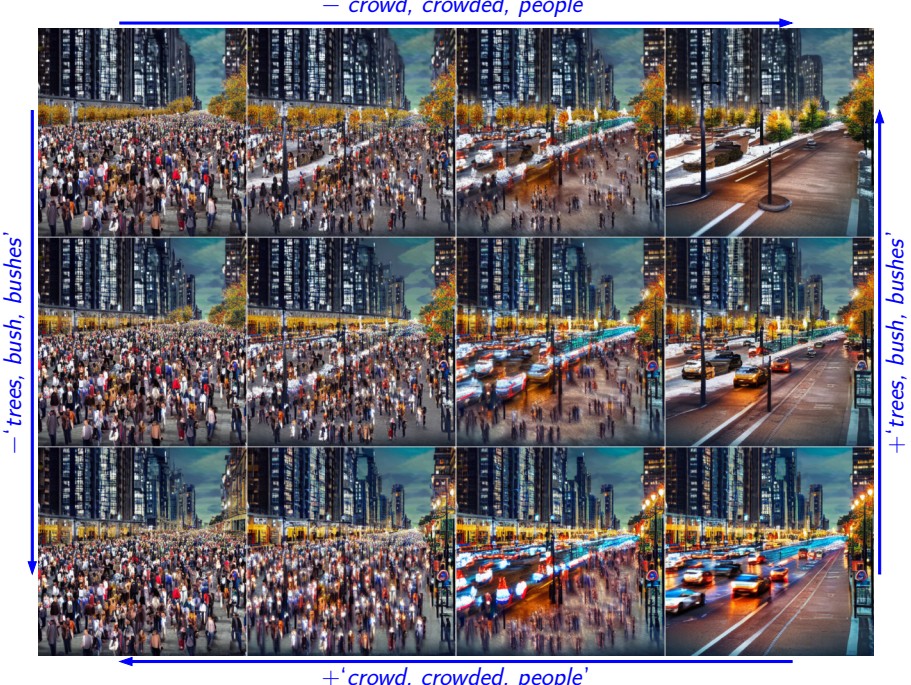

Figure 15: SEGA offers strong control over the semantic space and can gradually perform edits at the desired strength. All images generated from the same initial noise latent using the prompt 'a crowded boulevard'. Editing prompts denoted in blue and are gradually increased in strength from left to right and top to bottom. (Best viewed in color)

## D   Further Examples of Semantic Space's Properties

Here, we provide further examples of the properties of the semantic space created by SEGA (cf. Sec. 4). First, we revisit the uniqueness of guidance vectors. As shown in Fig. 3b these vectors may be calculated once and subsequently applied to other prompts. However, this is limited to the same initial noise latent and requires a similar image composition. We depict this difference in Fig. 8 where the guidance vector for '*glasses*' is successfully applied to three images and fails to add glasses on three other images. However, the unsuccessfully edited images on the right retain the same image fidelity showcasing that SEGA's guidance vectors do not introduce artifacts. Nonetheless, SEGA can easily incorporate glasses into these images by computing a dedicated guidance vector (cf. Sec. 4).

Next, we demonstrate that multiple concept vectors scale monotonically and independent of each other. Meaning that in a multi-conditioning scenario the magnitude of each concept can be targeted independently. We depict an example in Fig. 15. Starting from the original image in the top left corner, we remove one concept from the image following the x- or y-axis. In both cases, SEGA monotonically reduces the number of people or trees until the respective concept is removed entirely. Again, the rest of the image remains fairly consistent, especially the concept that is not targeted. Going even further, a similar level of control is also possible with an arbitrary mixture of applied concepts. Let us consider the second row of Figure 15, for example. The number of trees is kept at a medium level in that the row of trees on the left side of each image is always removed, but the larger tree on the right remains. While keeping the number of trees stable, we can still gradually remove the crowd at the same time.

## E   User Study

We subsequently provide further details on the empirical evaluation presented in Sec. 5. We attempted to label the generated images using a classifier trained on CelebA. However, despite the model achieving over 95% accuracy on the CelebA validation set, the accuracy for the synthetic images was

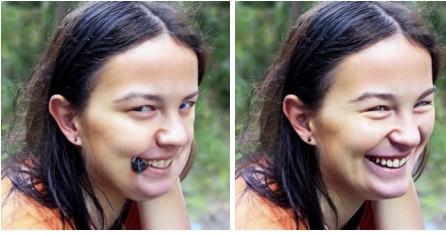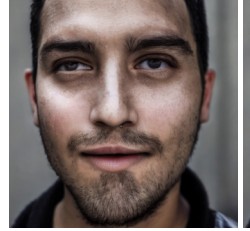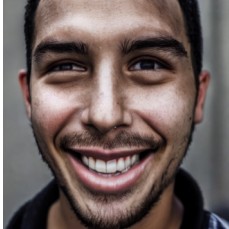

Figure 16: Examples of improved visual quality through SEGA editing. Uncanny artifacts in the original images (left) are attenuated or removed by the additional guidance in image regions edited using SEGA (right). The used edit is . '*smiling*'.

for some classes below 20%. Consequently, we opted to rely on human evaluators instead. For each of the 250 generated images we collected labels for all 10 attributes. Questions were posed in the form

Is the person wearing glasses?

The three answer options were:

- Yes
- No
- Cannot Tell

or in the case of the gender attribute:

- Male
- Female
- Cannot Tell

Each user was tasked with labeling a batch of 28 image/attribute pairs; 25 out of those were randomly sampled from our generated images and each batch contained 3 hand-selected images from CelebA as sanity check. If users labeled the 3 CelebA images incorrectly the batch was discarded and added back to the task pool. Each images/attribute combination was labeled by 3 different annotators resulting in annotator consensus if at least 2 selected the same label.

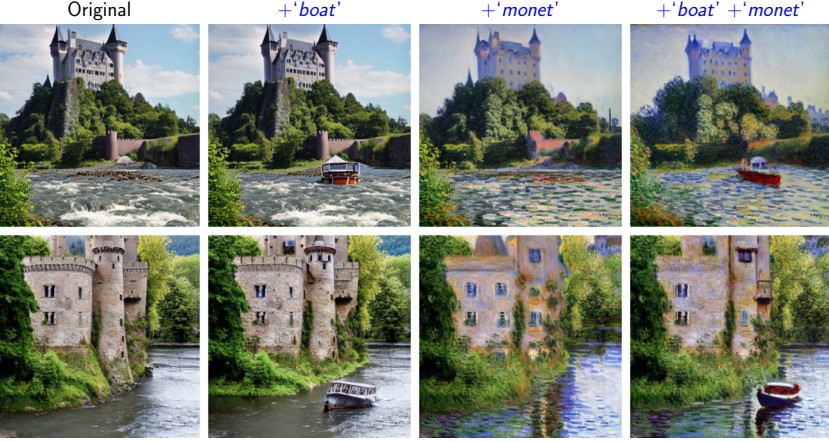

Figure 17: Robustness of SEGA across multiple tasks. Here style transfer and image composition are easily combined and executed simultaneously. The resulting image satisfies both tasks. All images generated using the prompt '*a castle next to a river*'. Top and bottom row use different seeds. (Best viewed in color)

To conduct our study we relied on Amazon Mechanical Turk where we set the following qualification requirements for our users: HIT Approval Rate over 95% and at least 1000 HITs approved. Annotators were fairly compensated according to Amazon MTurk guidelines. Users were paid $0.70 for a batch of 28 images at an average of roughly 5 minutes needed for the assignment.

In Fig. 16 we show examples of SEGA edits removing artifacts and improving overall *uncanny* image feel. Consequently, the evaluated image fidelity of the edited images is higher than of the originally generated ones.

## F  SEGA on various Architectures

As highlighted in the main body of this paper, SEGA can be integrated into any generative model using classifier-free guidance. Here we show further qualitative examples on various other architectures beyond Stable Diffusion. We implemented SEGA for two additional models: 1) Paella [26] which operates on a compressed and quantized latent space with iterative sampling over the distribution of predicted tokens per latent dimension. 2) IF [6] a publicly available pixel-level diffusion model based on the Imagen architecture [28].

## G  Comparisons to related methods

Subsequently, we compare SEGA to other methods for image editing.

### G.1  Direct Comparisons

We first compare against related methods for diffusion that allow for direct comparisons, i.e. the same seed and prompt will generate the exact same original image. All of these techniques have dedicated failure modes, i.e. use cases that these methods cannot accomplish. SEGA on the other behaves more robustly and successfully performs the desired edits in all scenarios.

#### G.1.1  Composable Diffusion

We first compare against Composable Diffusion [16]. In Fig 19 we can see that Composable Diffusion is also capable of conditioning on multiple concepts. However it makes more substantial changes than SEGA and produces artifacts when used with 3 or more concepts. Additionally, CompDiff performs significantly worse on image composition as shown in Fig. 20. For multiple of the applied

---

[6]https://github.com/deep-floyd/IF

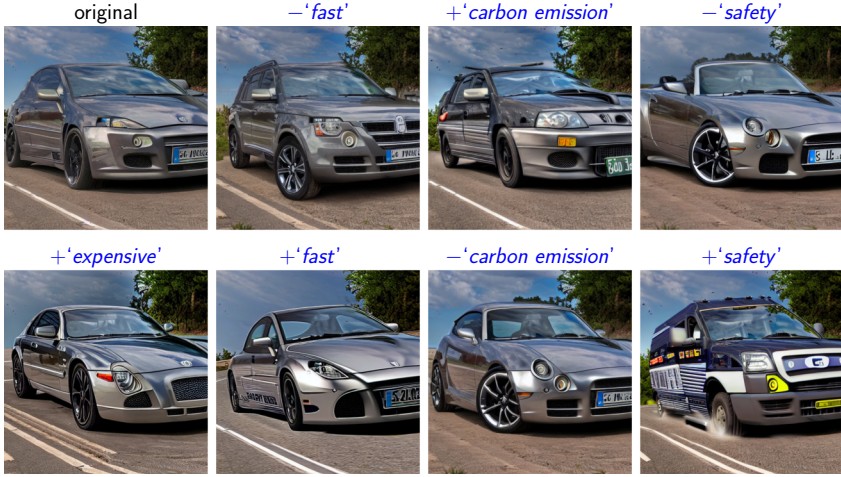

Figure 18: Probing the diffusion model for learned concepts. Guiding towards or away from more complex concepts may reveal the underlying representation of the concept in the model. All images generated using the prompt '*a picture of a car*'. (Best viewed in color)

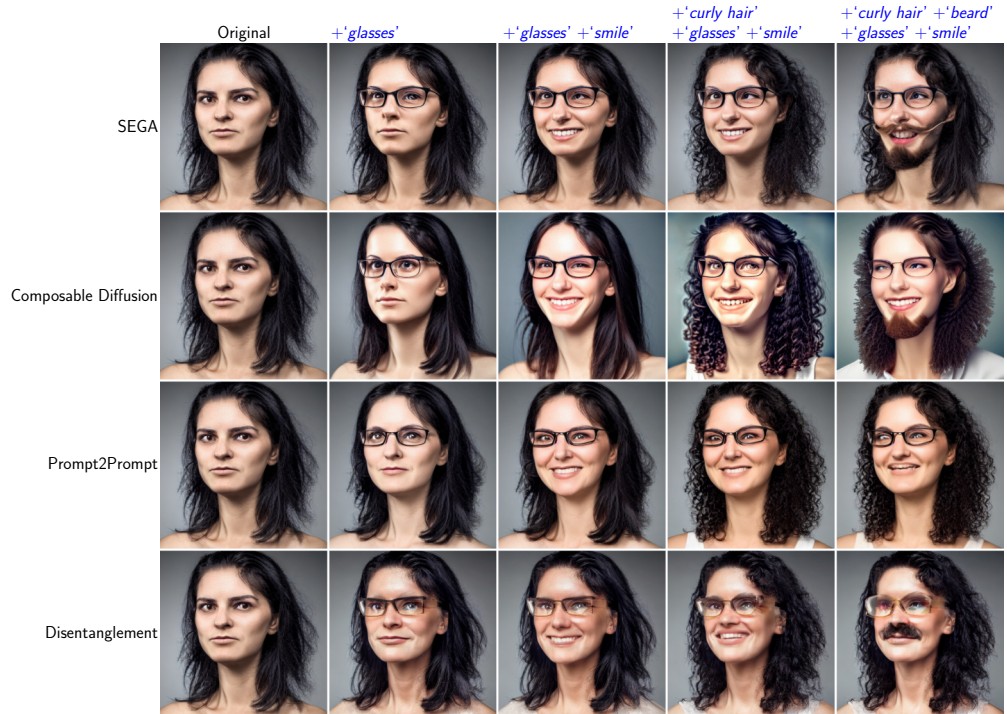

Figure 19: Comparison of multiple edits being performed simultaneously for various semantic diffusion approaches.

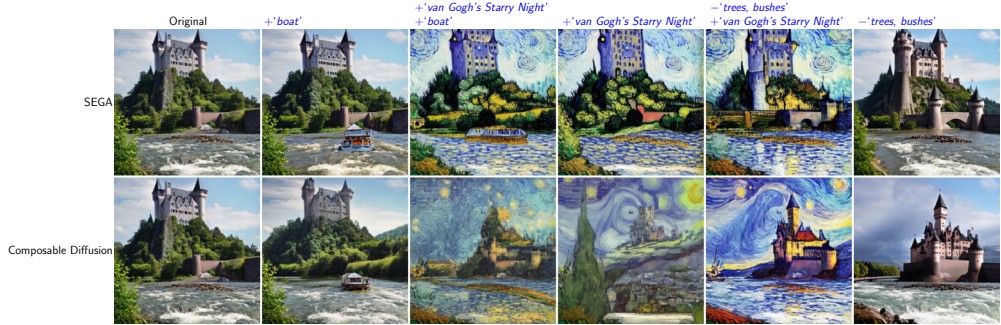

Figure 20: Comparison of SEGA against Composable Diffusion [16] on image composition. SEGA signficantly outperforms Composable Diffusion in terms of truthfulness to the original image and combination of multiple concepts.

and combined concepts the original image is no longer recognizable and SEGA also outperforms in terms of truthfulness of the targeted concepts and combinations thereof.

### G.1.2 Prompt2Prompt

Prompt2Prompt [8] generally produces edits of similar quality as SEGA. However, Prompt2Prompt is limited to one editing prompt. This leads to the some concepts being ignored when aiming to edit multiple ones simultaneously as shown in Fig 19.

### G.1.3 Disentanglement

The Disentanglement proposed by Wu et al. [35] performs similarly well to SEGA in applying multiple edits to faces (Fig 19). However, their method fails to realize small changes in an image as shown in Fig. 21. SEGA, on the other hand, faithfully executes the desired edit operation.

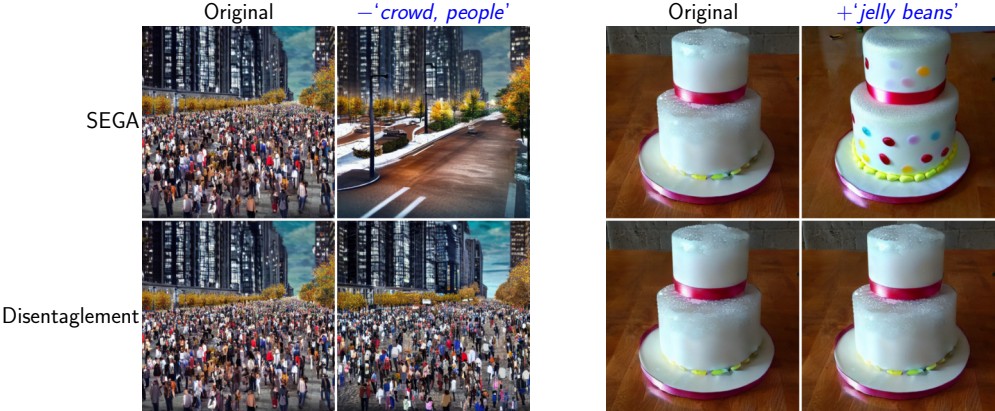

Figure 21: Comparison of SEGA against the approach of Wu et al. [35] for small edits. The Disentanglement makes no significant changes to the image, whereas SEGA faithfully executes the desired edit operation.

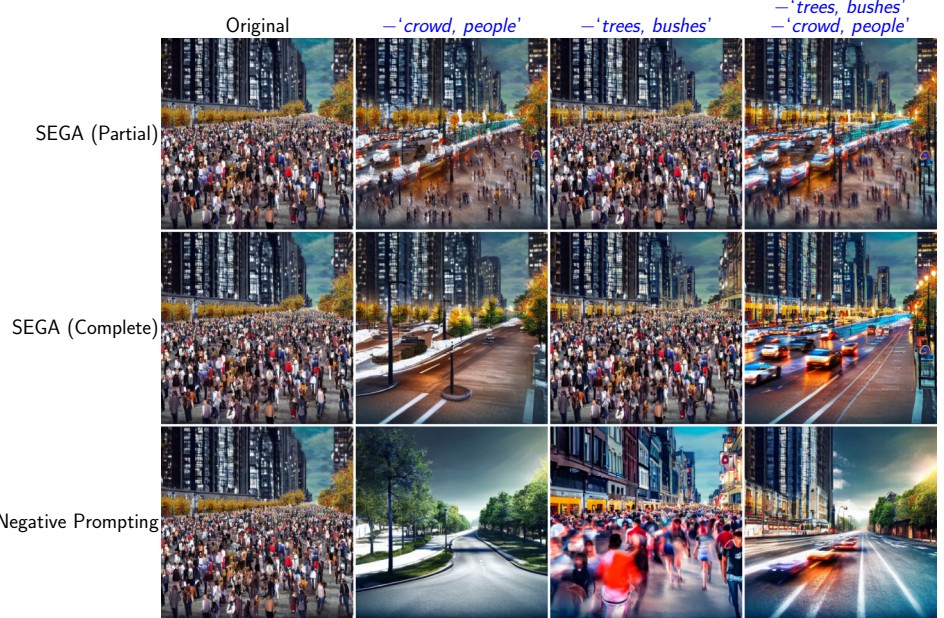

Figure 22: Comparison of SEGA against Negative Prompting. SEGA allows for dedicated control over the strength of the applied negative concept. Top row shows partial removal of the targeted concept with SEGA and middle row complete removal. Additionally, negative prompting breaks when used with multiple concepts simultaneously.

Additionally, the Disentanglement approach requires the optimization of a linear transformation vector, whereas SEGA can be applied ad hoc.

### G.1.4 Negative Prompting

SEGA offers multiple benefits over negative prompting as shown in Fig 22. 1) Negative prompting offers no control over the strength of the negative guidance term, whereas SEGA provides a dedicated semantic guidance scale. 2) This often leads to unnecessarily large changes to the image when using negative prompts with the image oftentimes being completely different. With SEGA, however, changes can be kept to a minimum. 3) Negative prompting does not allow for removal of separate concepts, instead multiple concepts have to be provided in a single prompt. This may lead to one of the concepts not being removed in the generated image. In contrast, SEGA provides one guidance term with dedicated hyper parameters for each concept.

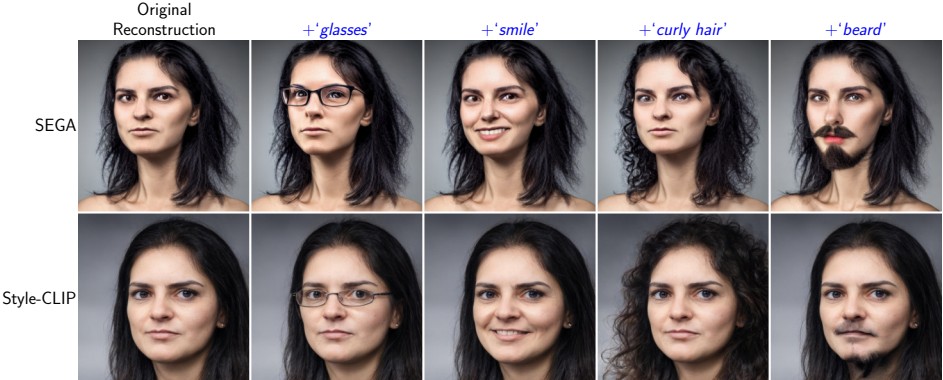

Figure 23: Comparison of SEGA against StyleCLIP.

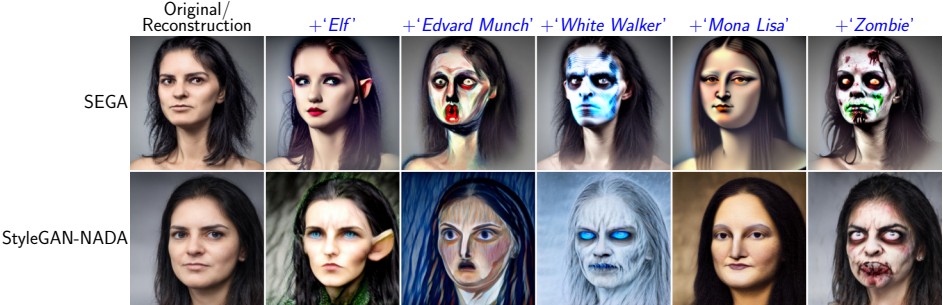

Figure 24: Comparison of SEGA against StyleGAN-NADA.

## G.2 Related Approaches

We also compare against two StyleGAN based approaches for semantic editing. Notably, we had to reconstruct the original image with StyleGAN which results in differences in the un-edited image.

### G.2.1 Style-CLIP

SEGA achieves edits of similar fidelity as StyleCLIP [22] as shown in Fig. 23. However, SEGA can be applied ad hoc and does not require the time consuming mapping of a prompt against a style space attribute as need for StyleCLIP. Furthermore SEGA works on a powerful diffusion model capable of generating diverse images beyond one domain, contrary to a dedicated StyleGAN for faces.

### G.2.2 StyleGAN-NADA

SEGA achieves edits of similar fidelity as StyleGAN-NADA [7] as shown in Fig. 24. However, SEGA can be computed on the fly and requires no tuning of the generative model as needed for StyleGAN-NADA. Furthermore SEGA works on a powerful diffusion model capable of generating diverse images beyond one domain, contrary to a dedicated StyleGAN for faces.

## H Further Qualitative Evaluation

Lastly, we provide further qualitative examples further highlighting the capabilities and use cases of SEGA.

The isolation of concept vectors (cf. Sec. 4) not only allows for multiple simultaneous edits of the same task but also arbitrary combinations. In Fig. 17 we simultaneously change the image composition and change the style. The resulting image is a faithful interpolation of both changes applied individually. This further highlights both the robustness of SEGA guidance vectors as well as their isolation.

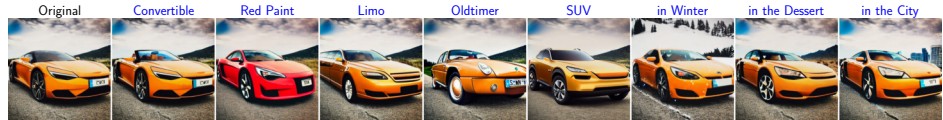

| Original | Convertible | Red Paint | Limo | Oldtimer | SUV | in Winter | in the Dessert | in the City |

(a) Examples of image editing with prompt 'an image of a car'. All images are generated from the same noise latent and text prompts. SEGA performs high quality local and global edits with minimal changes to other aspects of the image.

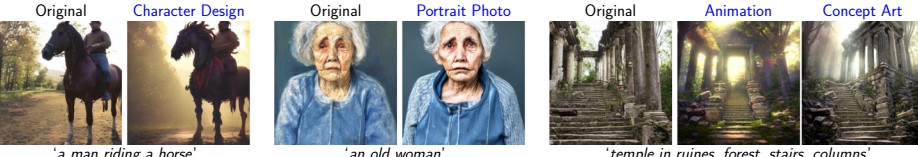

| Original | Character Design | Original | Portrait Photo | Original | Animation | Concept Art |

'a man riding a horse'   'an old woman'   'temple in ruins, forest, stairs, columns'

(b) Optimization of artistic conception. Guidance conditioned on average embedding of prompt targeting one type of imagery. The resulting images are of high quality, reflect the targeted artistic style, while staying true to the original composition.

Figure 25: Qualitative Examples of Semantic Guidance on various tasks and domains. (Best viewed in color)

In Fig. 25a, we can see various edits being performed on an image of a car. This showcases the range of potential changes feasible with SEGA, which include color changes, altering the vehicle type or surrounding scenery. In each scenario, the guidance vector inferred using SEGA accurately targets the respective image region and performs changes faithful to the editing prompt. All while making next to no changes to the irrelevant image portions.

An additional benefit of semantic guidance directly on noise estimates is its independence on the modality of the concept description. In the case of SD, the modality happens to be text in natural language, but SEGA can be applied to any conditional diffusion model. We explore this idea further with the goal of optimizing the overall artistic conception of generated images. Instead of defining the target concept with one text prompt, we directly use an abstract embedding, representing a particular type of imagery. To that end, we collected prompts known to produce high-quality results[7] for five different types of images in *portrait photography, animation, concept art, character design*, and *modern architecture*. The conditioning embedding for one style is calculated as the average over the embeddings of all collected prompts. Exemplary outputs are depicted in Fig. 25b. The results are of high quality and stay close to the original image but accurately reflect the targeted artistic direction beyond a single text prompt.

SEGA also contributes in uncovering how the underlying DM "interprets" more complex concepts and gives further insight into learned representations. We perform this kind of probing of the model in Fig. 18. For example, adding the concept '*carbon emissions*' to the generated car produces a seemingly much older vehicle with a presumably larger carbon footprint. Similarly, reducing '*safety*' yields a convertible with no roof and likely increased horsepower. Both these interpretations of the provided concepts with respect to cars are logically sound and provide valuable insights into the learned concepts of DMs. These experiments suggest a deeper natural language and image "understanding" that go beyond descriptive captions of images.

---

[7]Prompts taken from https://mpost.io/best-100-prompts

