# OpenReview forum: "SEGA: Instructing Text-to-Image Models using Semantic Guidance"
_NeurIPS.cc/2023/Conference — NeurIPS 2023 poster_

### Official Review · Reviewer_Lm7m · 2023-06-10

**Soundness:** 2 fair
**Presentation:** 3 good
**Contribution:** 2 fair
**Rating:** 3
**Confidence:** 5

**Summary:**

The paper introduces Semantic Guidance, a method that enables user control and interaction with text-to-image diffusion models to generate images that align with their intended semantics. The authors highlight the challenge that small changes in the input prompt can result in vastly different images. To address this issue, SEGA leverages the previously proposed composable diffusion approach to allow for fine-grained control over the diffusion process. The experiments showcase the effectiveness of SEGA mostly on human faces.

**Strengths:**

1. This paper studies an important problem, fine-grained control for diffusion models.

2. Experiments on human faces are extensive. Both automatic evaluation and human evaluation are conducted.

3. Results show that SEGA enables subtle edits on human faces, which disentangles the intended edits from factors that should remain untouched.


**Weaknesses:**

1. **Similarity to previous work:** The method is almost identical to Composable Diffusion [1], where different concepts are composed to manipulate the generated image.

2. **Limited scope of evaluation:** Most experiments are done on human faces. It is unclear if the method can be generalized to other object categories, e.g., the height of a building, the width of a bench, or moving an object left/right.

3. **Potential biases:** Several semantic edits target controversial concepts such as gender, hate, and violence. It's better to discuss what is the scope of evaluation here, e.g., what genders are considered and what remains unresolved.

[1] Compositional Visual Generation with Composable Diffusion Models. 2022

**Questions:**

1. What are the typical failure cases of the proposed method?

**Limitations:**

The limitation of the method and evaluation setting needs to be discussed.

---

> ### Author Rebuttal · Authors · 2023-08-09
>
> We thank the reviewer for their valuable feedback and suggestions. We will address each of the stated weaknesses and questions individually.
>
> ### W1: Similarity to previous work
> We respectfully disagree with the reviewer that our work is almost identical to Composable Diffusion. We agree that there are some similarities between the approaches, which is why we discuss Composable Diffusion in our related work.
> As highlighted by the other reviewers, we argue that SEGA makes a relevant contribution by identifying concept encoding dimensions of the noise estimates, as well as other manipulation techniques yielding fine-grained semantic control.
> The improvements provided by SEGA are also clearly highlighted in the comparison user study we provide in the general comment above. When compared to ComposableDiffusion SEGA offers better success rates and editing quality for tasks related to multi-conditioning and small changes while providing preferred edits in cases of style transfer and removal at comparable success rates.
>
> ### W2: Limited scope of evaluation
> We would like to point out that the paper does provide an extensive, empirical study beyond human faces, as presented in Tab. 3. Additionally, we believe that the user study presented in the general comment above, which compares SEGAs performance to other techniques on a general set of editing tasks, addresses the limitation in evaluation stated by the reviewer.
>
>
> ### W3: Potential biases
> We agree with the reviewer that the paper would benefit from a more nuanced discussion of ethical and societal implications. Consequently, we adjusted the paper to better reflect the following biases and limitations:
> 1. We acknowledge that the editing of perceived gender in the paper may exhibit potential biases due to the following factors. Due to the limitation of Stable Diffusion’s learned representation of gender, we restrict ourselves to binary labels, although gender expression cannot be ascribed to two distinct categories. Furthermore, does Stable Diffusion contain severe biases in its attribution of gender and correlated features that may further reinforce pre-existing stereotypes. Since editing methods, in general, and SEGA in particular, rely on these representations of the underlying model, they will inevitably inherit them for related editing operations.
> 2. The results presented on the suppression of inappropriate content are based on the definitions of inappropriateness introduced by the authors of Safe Latent Diffusion. We agree with the authors that what is considered inappropriate imagery may differ based on context, setting, cultural and social predisposition, and individual factors and is highly subjective overall. Again using the representations of the model itself assumes a definition of inappropriateness that may appear biased to some
> 3. While we believe SEGA to be an amazing tool promoting creating artwork in an interactive process, we want to acknowledge the group of artists feeling threatened by the wide accessibility of these models. Additionally, we want to highlight the ethical implications of using web-scraped data for the training of generative models. Data that contains the artwork of people who have not provided explicit consent for the usage in ML training.
>
> ### Q1: Typical failure cases
> As with all editing methods, a key failure case is attempting edits for concepts the underlying model has no clear representations of. While these cases are limited in a generalistic model such as stable diffusion, examples can include unknown artists or highly specific domains such as medical imaging.
> Additionally, targeting only one of several same objects (e.g., two people) is challenging, but also not impossible. We believe that further improvements to the masking process (c.f. Fig. 24 in the general response PDF) can be a viable solution for this issue which we are currently investigating for future work.

---

### Official Review · Reviewer_ZrN7 · 2023-06-16

**Soundness:** 4 excellent
**Presentation:** 3 good
**Contribution:** 3 good
**Rating:** 6
**Confidence:** 5

**Summary:**

This paper proposes a method for attribute editing of the images generated with Text-to-Image Diffusion models (T2I-DM). This is done in a manner similar to CFG, where extra edit prompts are passed alongside the text prompt to generate the image. Noise predicted from the edit prompt in the denoising process guides the editing of the generated image. The properties of SEGA are extensively reported with qualitative results and on multiple T2I-DMs.

**Strengths:**

1. The key idea is simple, well-motivated, and theoretically backed.
2. The four semantic properties of the method are extensively studied and experimentally backed.
3. Human evaluation results prove the effectiveness of the method.


**Weaknesses:**

1. In Table 1, I suppose the 250 images could or could not contain an attribute (for eg. smile). In that case, the results have been reported by adding smile to 146 images which did not contain it and removing smile from the 93 images that previously had a smiling face. If so, the same protocol could be followed for all other attributes. However, the results for negative attributes are reported only for 3/9 attributes (excluding gender).

2. It would complete the paper if a user preference study against competitive methods such as Wu et al. [1] are also reported.

[1] Wu et al. Uncovering the Disentanglement Capability in Text-to-Image Diffusion Models. https://arxiv.org/abs/2212.08698


**Questions:**

Insights about the hyperparameter choices in the supplementary could be presented better. The conclusion statements in each case are missing. What aspect of the generation (image quality, details etc) are controlled by each of the hyperparameter? Perhaps a summary of the hyperparameter analysis could be made part of the main paper, instead of Fig. 6 and its analysis.

**Limitations:**

Limitations have been addressed adequately.

---

> ### Author Rebuttal · Authors · 2023-08-09
>
> We thank reviewer ZrN7 for their feedback and subsequently address each stated weakness and question.
>
> ### W1: Additional results for negative attributes
> While we believe the presented attributes for negative guidance to be representative of SEGA’s capabilities, we agree that the results on negative guidance for the additional attributes would provide a more holistic view. Consequently, we performed the respective experiments with the following results:
>
> |   Attribute   |   Samples   |   Consensus (%)   |   Success (%)   |
> | --- | --- | --- | --- |
> |   No gray hair   |   52   |   94.55   |   82.69   |
> |   No curly hair   |   50   |   98.04   |   98.00   |
> |   No bald   |   18   |   94.74   |   83.33   |
> |   No makeup   |   21   |   100.0   |   95.24   |
> |   No hat   |   31   |   100.0   |   90.32   |
> |   No bangs   |   38   |   95.00   |   89.47   |
> |   **Total** |  **210**   | **96.77** |  **90.00**  |
>
>
> The overall success rates remains very similar to the results reported in the paper, thus providing further evidence of the capabilities of SEGA.
>
> ### W2: Comparative user study
> We conducted a user preference study comparing SEGA to Prompt2Prompt, Composable Diffusion, and Disentanglement. In summary, the results demonstrate that SEGA generally outperforms all these methods, with more details being provided in the general response above.
>
> ### Q1: Hyperparameter analysis
> We provide further details on each hyperparameter in a general comment above. Given the additional page provided for the camera-ready version, we would include this analysis in the main body of the paper with no need for removing other content.

---

### Official Review · Reviewer_q8hy · 2023-07-06

**Soundness:** 3 good
**Presentation:** 4 excellent
**Contribution:** 3 good
**Rating:** 7
**Confidence:** 4

**Summary:**

The paper proses a novel approach called Semantic Guidance (SEGA) to edit an image using text only and without fine-tuning the T2I model. It does this by manipulating the noise estimate in the T2I diffusion model. The space of sparse noise-estimate vectors is referred to as the semantic space. The paper justifies various desirable properties of SEGA such as altering fine-grained output details, architecture agnostic (as long as it uses classifier-free guidance), works on both pixel and latent space, no T2I fine-tuning, supports editing of multiple attributes (addition or deletion). Guidance vector is unique per concept and thus can be pre-computed. The value of semantic guidance vector controls the magnitude of semantic concept (e.g. degree of smile). Different concepts/modifiers typically affect different regions of the noise estimate.  Experiments in the paper has good qualitative results as well as positive human evaluation on  2 data sets.

**Strengths:**

S1) Clarity

S2) Simplicity, elegance, effectiveness, generality of the proposed approach

S3) Several useful insights regarding the semantic vector

S4) Good qualitative results and human evaluation


**Weaknesses:**

W1) Quantitative evaluation on a large data set. E.g. face attribute detection could be used to do large scale evaluation

W2) Not sure how effective SEGA is for more complex manipulations involving multiple objects in the same image e.g. controlling the color of 2 objects independently using referring expressions.


**Questions:**

Please see list of weakness

**Limitations:**

Yes

---

> ### Author Rebuttal · Authors · 2023-08-09
>
> We thank the reviewer for their great feedback and address each weakness separately.
> ### W1: Large-scale evaluation w/ face attribute detection
> In line with the reviewer’s suggestion, we initially attempted to use face attribute detection for large-scale evaluation. As outlined in the Appendix, however, the drastic shift between labeled, real images and Stable Diffusion generated synthetic images made an automatic evaluation infeasible. Presumably, also due to the oftentimes generated uncanny artifacts in facial images (cf. Fig.25  in general response PDF). In fact, we trained a classifier on CelebA, achieving over 95% on a held-out validation set that we attempted to use for labeling images edited with SEGA. However, an investigation of this classifier’s performance for synthetic images resulted in less than 20% accuracy for glasses, for example, rendering such a method impractical for the evaluation of SEGA.
>
> ### W2: Complex manipulations
> Complex manipulations targeting different objects independently are indeed possible with SEGA. As shown in the general response PDF, SEGA’s threshold λ identifies implicit masks of regions in an image corresponding to the desired edit. Consequently, manipulating two concepts independently requires a set of edit expressions and thresholds that separates the objects accordingly. We agree with the reviewer that this is a promising research direction and are currently investigating such complex manipulations for future work. We extended the discussion in the conclusion accordingly.

---

> > ### Comment · Reviewer_q8hy · 2023-08-17
> >
> > Thanks to the authors for addressing my concerns.

---

### Official Review · Reviewer_i1VN · 2023-07-06

**Soundness:** 3 good
**Presentation:** 3 good
**Contribution:** 4 excellent
**Rating:** 6
**Confidence:** 3

**Summary:**

This work proposes SEGA (semantic guidance) for text-to-image diffusion models. It is a architecture-agnostic technique and is only applied during the sampling process. In addition to classifier-free guidance, a semantic guidance is applied that pushes noise prediction along the direction of a specific semantic concept.

**Strengths:**

The technique is simple and intuitive. It achieves fine-grained semantic control of image generation via noise-estimate manipulation, therefore an easier process than image-editing models. The results look interesting.


**Weaknesses:**

The human study only evaluates on whether a label is present/absent. No study is done on the quality of images. For example, adding 'glasses' on an image hopefully will not alter the other attributes of the image, i.e. the person still looks the same, the facial expression remains the same, etc. And in addition, the image quality also should not drop significantly. These can be done via user study, or image quality can be measured by inception scores or FID. (I think at least some measure of image quality comparison with the unaltered images is needed.)

**Questions:**

Choices of $c_e$ presented in the paper are all single-word, precise semantic concepts. You tried combining multiple concepts via Eq.10. Have you thought about making $c_e$ a composite concept?

**Limitations:**

The authors have discussed the limitations and broader impact.

---

> ### Author Rebuttal · Authors · 2023-08-09
>
> We thank reviewer i1VN for their input and questions. Below we address each weakness and question individually.
>
> ### W1: Image quality measurement.
> We agree with the reviewer that a measure of image quality comparison with unaltered images will improve the paper substantially and conducted a respective experiment. In summary, these additional results give no indication of quality degradation due to SEGA editing. In fact, the scores improve significantly for the edited images.
>
> As suggested, we calculated FID scores of generated facial images against FFHQ for the original images and edited ones generated for the experiments in Tab 1 & Tab 2.
> The FID score for the original images are considerably low at 117.73, which can be attributed to small artifacts often present in facial images generated by Stable Diffusion.
> Editing images with SEGA significantly improved their quality resulting in an FID score on FFHQ of 59.86. Upon further investigation, we observed that the additional guidance signal for a dedicated portion of the faces frequently removed uncanny artifacts, resulting in overall better quality. We include exemplary images highlighting this behavior in the general response PDF (Fig. 25).
> Additionally, we conducted a preference selection user study, including comparisons of edit quality. Further details are provided in a general comment above.
>
> ### Q1) Composite concepts
> SEGA may also be used with a composite concept prompt. In fact, the experiment conducted on suppressing the generation of inappropriate content (Tab. 3) uses the composite concept defined by SLD to describe unwanted material:
> ‘hate, harassment, violence, suffering, humiliation, harm, suicide, sexual, nudity, bodily fluids, blood, obscene gestures, illegal activity, drug use, theft, vandalism, weapons, child abuse, brutality, cruelty’
> As demonstrated by the classification results over different categories, SEGA actively suppresses inappropriate material for each of these categories with only one edit prompt containing multiple concepts.

---

> > ### Comment · Reviewer_i1VN · 2023-08-20
> > **Response**
> >
> > Thanks to the authors for the rebuttal and the added evaluation and information. It is interesting (and counter-intuitive to me) that SEGA editing improves FID significantly. The observation that "the additional guidance signal for a dedicated portion of the faces frequently removed uncanny artifacts" is an interesting one, and it would be great if this can be reasoned for as well. Overall I maintain my positive rating of weak accept.

---

### Official Review · Reviewer_2nZZ · 2023-07-11

**Soundness:** 3 good
**Presentation:** 3 good
**Contribution:** 3 good
**Rating:** 5
**Confidence:** 5

**Summary:**

This paper analyzes classifier guidance in a more detailed way. The main idea is to tie the direction of guidance and the semantic, so that multiple guidance can be used. This idea is simple and effective. The paper also presents many small tricks to improve the performance.

SEGA is evaluated on the face manipulation task and some qualitative benchmarks with a high successful rate.

**Strengths:**

1. Classifier-free guidance is a very interesting topic in diffusion models, which is the key to improve the performance but neglected by many theoretical papers.
2. The paper tests their method on SD, Paella and DeepFloyd-IF, exhibiting its generality.

**Weaknesses:**

1. The paper presents a solid analysis, but not deep enough for classifier-free guidance. In my opinion, two things are very interesting:
(1) What is the range of validity for the linearity of semantics? In word2vec, although the story of linearity is very appealing, but it is not true for many words. In diffusion model, there should be also a range to keep the linearity, which is very important for extending classifier-free guidance.
(2) Figure 2 shows only the top 1-5% values are enough to guide the generation. This is an important finding, but it seems like a total statistic from multiple images. In a specific images, where are the top values distributed spatially? What is the relation between these values and initial noise?
2. $\mu$ is used to select the dims with the largest absolute values, which is the largest different between SEGA and usual CFG. However, more discussions about the motivation and quantitative results are needed. The importance of other tricks, e.g. warmup and momentum, are also unknown.
3. The experiments are mainly limited to face-related and qualitative cases. I still think it could be better to compare SEGA with other baselines, e.g. prompt2prompt.


**Questions:**

See weakness.

**Limitations:**

See weakness.

---

> ### Author Rebuttal · Authors · 2023-08-09
>
> We thank the reviewer for their valuable feedback and will address each weakness individually.
>
> ### W1: Analysis of classifier-free guidance
> 1. **Linearity of Semantics.**
> We would like to clarify that SEGA concepts scale monotonically and not necessarily linearly, as the space of noise estimates and the respective concepts in image space do not have a well-defined distance metric. Consequently, it is difficult to directly compare word2vec, for example.
> We agree with the reviewer that this question is important for a deeper understanding of classifier-free guidance. In general, scaling reaches its limit when the numerical values are pushed out of their well-defined bounds, leading to image artifacts. We observed SEGA to be less susceptible to this issue than CFG, producing less artifacts at high guidance scales. We added a more detailed discussion on the limits of scaling to the paper.
>
> 2. **Spatial distribution of SEGA dimensions**
> The spatial distributions of activating SEGA dimensions above the threshold are closely correlated to the edited image region. We provide an example in the general response PDF above (Fig. 24). In the example of adding glasses to an image, these dimensions are concentrated on the location of the glasses in the image. Consequently, the spatial distribution changes based on the currently generated image and the semantically grounded location of the targeted edit. We believe these findings to be insightful in understanding classifier-free guidance and adjusted the paper to provide further clarification.
>
> ### W2: Motivation of μ and importance of hyperparameters
> The largest absolute difference is computed between the noise estimate of the edit concept and the unconditioned concept. In diffusion literature, this can be considered as the gradient of an implicit classifier steering the generation toward the edit concept. Consequently, the dimensions with the largest absolute value corresponds to the highest information density from this implicit classifier. As discussed above, these dimensions produce an implicit mask of the area to be edited while remaining semantically grounded in the currently generated image.
> We included a more thorough discussion of this motivation in the paper.
> Furthermore, we added a more detailed discussion of the importance of each hyperparameter – including warmup and momentum – to the paper as outlined in the general comment above.
>
> ### W3: Comparisons to other baselines
> We conducted a user preference study comparing SEGA to Prompt2Prompt, Composable Diffusion, and Disentanglement. In summary, the results demonstrate that SEGA generally outperforms all these methods with more details being provided in the general response above.

---

### Author Rebuttal · Authors · 2023-08-09

We thank all reviewers for their valuable feedback and suggestions, which we have taken under careful consideration to improve the paper.
## Comparison
As suggested by the reviewers, we conducted randomized user studies comparing the capabilities of SEGA with related methods. Namely: Prompt2Prompt (P2P) [8], Disentanglement [35], and Composable Diffusion (CompDiff) [16].
To that extent, we considered tasks from four types of manipulation categories, reflecting a broad range of operations.
1) Composition of multiple edits similar to the experiment reported in Tab. 1 & 2.
2) Minor changes (e.g., change of pizza toppings, turning a car into a convertible, etc. )
3) Various style transfers
4) Removing a set of specific objects from a scene (e.g., trees from a view of a landscape)

For each method, we selected the best-performing hyperparameters on a set of seeds < 100 and subsequently generated non-cherry-picked edits on a fixed test set (i.e., first applicable seeds >= 100).
We subsequently asked users to assess the success of an edit based on the presence/absence of the target attribute(s). Success rates per category are as follows:

|Method|Multi-Condition (%)|Minor Changes (%)|Style Transfer (%)|Remove (%)
|--|--|--|--|--
|SEGA|80|90|90|30
|Composable Diffusion|35|72|100|35
|Prompt2Prompt|35|68|65|5|
|Disentanglement|35|65|65|0|

We would like to note that the comparatively low scores on removal tasks result from the methods often reducing the presence of the targeted objects but not eliminating all of them in some cases. However, this peculiarity of the evaluation affects all methods similarly and does not influence the comparability of method capabilities.

These results demonstrate that SEGA clearly outperforms P2P and Disentanglement on all examined editing tasks. Compared to CompDiff, SEGA again has significantly higher success rates for Multi-Conditioning and Minor changes while achieving comparable performance for style transfer and object removal.

Lastly, we investigated user preference with respect to the degree of changes made by each method. To that end, we looked at pair-wise comparisons of samples that both methods edited successfully and asked users to asses the similarity with the original image.

Compared to Disentanglement, users strongly prefer SEGA results for 83.33% (vs. 13.33%) of samples, indicating the generally better performance of SEGA in terms of success rate and edit preference.

When compared to CompDiff users also strongly prefer the SEGA-edited images (80% vs 15% overall). Consequently, SEGA offers better success rates and editing quality for tasks related to multi-conditioning and small changes while providing preferred edits in cases of style transfer and removal at comparable success rates (83.33% vs. 8.33% preference for style and 100% vs. 0% preference for removal).

Lastly, comparing SEGA to P2P, our method offers better success rates across all categories. However, users preferred the edits of P2p for minor changes and style transfer in cases where both methods performed successful edits (17.86% vs. 82.14%). Nonetheless, SEGA remains more versatile in general while also providing methodological benefits, such as the fact that SEGA edit instructions can be provided in an independent manner without dedicated connections to certain tokens of the original prompt required for P2P. Additionally, we argue that the granularity of edits on a single sample can easily be adjusted to the user’s preferences using the sophisticated hyperparameters provided by SEGA.

[8] P2P arXiv:2208.01626

[16] CompDiff arXiv:2206.01714

[35] Disentanglement arxiv:2212.08698

## Hyperparameter analysis
As suggested by the reviewers, we added a more detailed discussion on the influence of each individual hyper-parameter, which can be summarized as follows:

**Scale s_e**. The magnitude of the expression of a concept in the edited image scales monotonically with s_e. Overall, the SEGA scale behaves very robustly for a larger range of values. On the one hand, values of around 3 or 4 are sufficient for delicate changes. On the other hand, s_e is less susceptible to producing image artifacts for high values and can often be scaled to values of 20+ w/o quality degradation if required for the desired expression of the concept.

**Threshold λ**. SEGA automatically identifies the relevant regions of an image for each particular set of instructions and the original image in each diffusion step. Image regions outside these implicit masks (cf. Rebuttal PDF Fig. 1) will not be altered by SEGA, leading to minimal changes. λ corresponds to the portion of the image left untouched by SEGA. Consequently, higher values of λ close to 1 allow for minor alterations, whereas lower values may affect larger areas of the image. Depending on the edit, λ >= 0.95 will be sufficient for most edits, such as small changes, whereas alterations of the entire image, like style transfer, usually require values between 0.8 and 0.9.

**Warmup δ**. In line with previous research, we have observed that the overall image composition is largely generated in early diffusion steps, with later ones only refining smaller details (cf. Rebuttal PDF Fig. 26).
Consequently, the number of warmup steps may be used to steer the granularity of compositional changes. Higher values of δ >= 15 will mostly change details and not composition (e.g., style transfer). Whereas strong compositional editing will require smaller values, i.e., δ >= 5.

**Momentum**. Contrary to previous hyper-parameters, momentum is more of an optional instrument to further refine the quality of the generated image. Most edits produce satisfactory results without the use of momentum, but image fidelity can further improve when used. Importantly, its main use case is in combination with warmup. Since momentum is already gathered during the warmup period, higher momentum facilitates higher warmup periods in combination with more radical changes.

---

### Decision · Program_Chairs · 2023-09-21

**Decision:**

Accept (poster)

**Comment:**

The AC has carefully read the paper, reviews, author response, and discussions. This paper proposes semantic guidance that pushes noise prediction along the direction of a specific semantic concept. The idea is simple and interesting. 3 out of 4 reviewers are positive. The AC agrees with the reviewers that the paper presented a simple yet effective approach for attributing editing in the diffusion process, which will be of wide interest to the ML community. And thus recommend acceptance. However, the authors are strongly encouraged to follow reviews to polish the paper, for example, adding more comparisons with related works.